# A single-cell atlas of pig gastrulation as a resource for comparative embryology

Luke Simpson[1], Andrew Strange [1], Doris Klisch[1], Sophie Kraunsoe [1,4], Takuya Azami[2], Daniel Goszczynski[1], Triet Le Minh[1], Benjamin Planells[1], Nadine Holmes[3], Fei Sang[3], Sonal Henson [3], Matthew Loose [3], Jennifer Nichols[2] & Ramiro Alberio [1] ✉

Cell-fate decisions during mammalian gastrulation are poorly understood outside of rodent embryos. The embryonic disc of pig embryos mirrors humans, making them a useful proxy for studying gastrulation. Here we present a single-cell transcriptomic atlas of pig gastrulation, revealing cell-fate emergence dynamics, as well as conserved and divergent gene programs governing early porcine, primate, and murine development. We highlight heterochronicity in extraembryonic cell-types, despite the broad conservation of cell-type-specific transcriptional programs. We apply these findings in combination with functional investigations, to outline conserved spatial, molecular, and temporal events during definitive endoderm specification. We find early FOXA2 + /TBXT- embryonic disc cells directly form definitive endoderm, contrasting later-emerging FOXA2/TBXT+ node/notochord progenitors. Unlike mesoderm, none of these progenitors undergo epithelial-to-mesenchymal transition. Endoderm/Node fate hinges on balanced WNT and hypoblast-derived NODAL, which is extinguished upon endodermal differentiation. These findings emphasise the interplay between temporal and topological signalling in fate determination during gastrulation.

The blueprint of the mammalian body plan is laid down during gastrulation, a fundamental process of embryonic morphogenesis that ends with the establishment of the three basic germ layers. Gastrulation can be sub-divided into "primary gastrulation" describing early germ-layer formation events prior to the formation of the node, and "secondary gastrulation" encompassing convergent extension, the onset of neurulation and somitogenesis[1]. The unfolding of these processes has been mapped using single-cell transcriptomics in the mouse[2], rabbit[3,4], nonhuman primates[5,6], and partially in humans[7]. Cross-species analyses have identified broadly conserved and divergent features of major lineage emergence, however, detailed investigations of "primary gastrulation" are limited due to the scarcity of cells in these datasets. Furthermore, validation of transcriptomic observations using embryos is limited by the lack of specimens in non-human primates and humans. To address this, here we present a high-resolution single-cell transcriptomic atlas of pig gastrulation and early organogenesis, comprised of 91,232 cells from 62 complete pig embryos collected between embryonic days (E) 11.5 to 15 (equivalent to Carnegie stages, CS, 6 to 10). The pig embryo, like most other mammals, forms a flat embryonic disc (ED) before the onset of gastrulation and represents an accessible species for functional investigations[8]. Importantly, the pig is a valuable model for biomedical research and is increasingly being utilized for the development of transplantable organs for humans[9–11].

Here, we used this comprehensive dataset to shed light on the salient features of gastrulation in mammals. By performing cross-

[1]School of Biosciences, University of Nottingham, Sutton Bonington Campus, Nottingham LE12 5RD, UK. [2]MRC Human Genetics Unit, Institute of Genetics and Cancer, University of Edinburgh, Western General Hospital, Crewe Road South, Edinburgh EH4 2XU, UK. [3]School of Life Sciences, University of Nottingham, Nottingham NG7 2RD, UK. [4]Present address: The Francis Crick Institute, 1 Midland Road, London NW1 1AT, UK. ✉e-mail: Ramiro.alberio@nottingham.ac.uk

species comparisons we uncover heterochronic differences in the development of extra-embryonic cell types. Despite variability in differentiation dynamics and pathways regulating cell behaviour, there is broad conservation in cell type-specific programs across pigs, primates and mice. We focussed on the long-standing question of how the definitive endoderm (DE) emerges during mammalian gastrulation. Despite the evidence of mesendodermal progenitors in invertebrates, fish and chick[12–15], recent studies in mice demonstrated that epiblast cells give rise to DE independent of mesoderm[16,17]. However, evidence from studies using mouse and human embryonic stem cells (hESC) suggests that a common mesendodermal progenitor may also exist in mammals[18–21]. We combined transcriptomic analysis and embryo imaging to show that soon after the first mesodermal cells appear in the posterior epiblast a group of ED disc cells expressing FOXA2+ delaminate to give rise to DE, these cells differ from later FOXA2/TBXT+ cells which give rise to the node/notochord. Further, both cell types form via a mechanism independent of mesoderm and do not undergo epithelial-to-mesenchymal transition (EMT). Further, functional validations

using in vitro differentiation of pluripotent pig embryonic disc stem cells (EDSCs) and hESC[22,23] show that a balance of WNT and Activin/NODAL signalling are critical to the acquisition of the endoderm fate. Together, our findings indicate that the temporal dynamics and spatial localisation of WNT, originating from the primitive streak, coupled with hypoblast-derived NODAL play critical roles in orchestrating primary gastrulation in mammals.

## Results

### Single-cell transcriptome of gastrulation and organogenesis

To investigate cellular diversification during gastrulation and early organogenesis in bilaminar disc embryos, we obtained scRNA-seq profiles from 23 pooled samples encompassing 62 pig embryos, collected at twelve-hour intervals between E11.5 and E15 using the 10X chromium platform (Fig. 1a; Supplementary Fig. 1). The dataset includes early streak up to 10-somites stage, equivalent to Carnegie stages (CS) 6 to 10 (Fig. 1b). Transcriptomes of 91,232 cells passed quality controls, with a median of 3,221 genes detected per cell (see Methods; Supplementary Fig. 1a–c). Known cell-type markers and

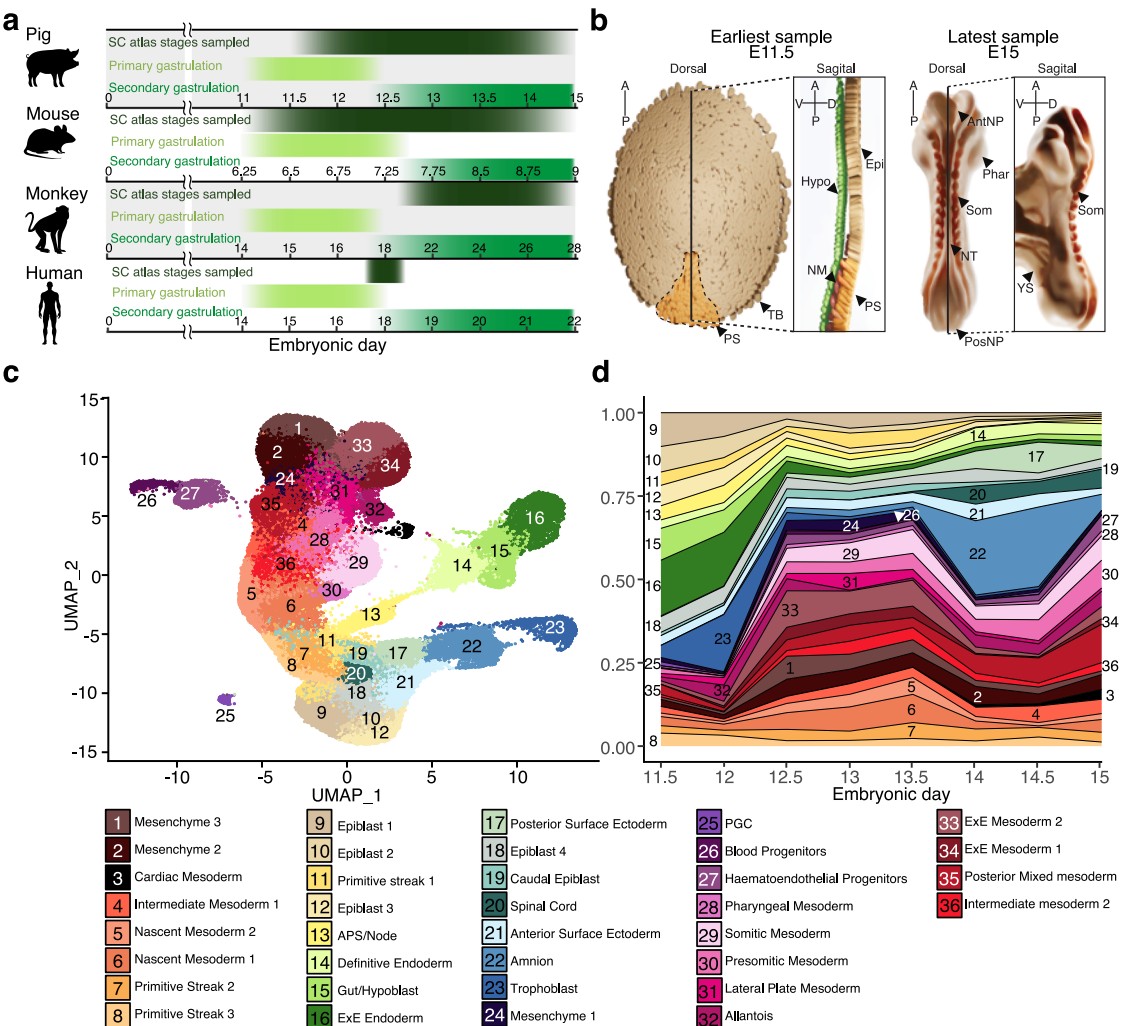

**Fig. 1 | Overview of the pig single-cell atlas. a** Onset and duration of primary and secondary gastrulation in humans, mice, pigs and monkeys. Timepoints where high-resolution single-cell datasets are available, are marked for each species[2,6,28] as well as the time points covered in this atlas. Numbers indicate embryonic day. **b** Diagrammatic representations of the earliest and latest embryo samples in this dataset with visible embryonic structures/cell types labelled. Epi Epiblast, PS Primitive streak, NM Nascent mesoderm, Hyp Hypoblast, TB Trophoblast, NT Neural tube, Phar Pharyngeal arches, Som Somites, PosNP Posterior neuropore, AntNP

Anterior neuropore, YS yolk sac. **c** Uniform manifold approximation and projection (UMAP) plot showing atlas cells (91,232 cells, 23 biologically independent samples). Cells are coloured by their cell-type annotation and numbered according to the same legend as d below. **d** Stacked area plot showing the fraction of each cell type at each time point, a progressive increase in cell-type complexity can be seen across time points with mesodermal cell type diversification preceding that of ectoderm. APS Anterior Primitive Streak, ExE extra-embryonic, PS Primitive streak, NM Nascent mesoderm, HE Hematoendothelial, PGC Primordial germ cells.

unbiased clustering of all samples (See Supplementary Data 1; Supplementary Fig. 1e) were used to identify 36 major cell populations and subsequently, pig cell-type marker genes (see Methods, Fig. 1c; Supplementary Data 2, 3). Early embryonic cell types, such as epiblast and primitive streak (PS) cells, decreased in number over time concomitant with their differentiation (Fig. 1d). Mesoderm and DE progenitors were present as early as E11.5 suggesting that gastrulation may commence before the morphological changes due to A-P patterning become visually apparent. Most mesoderm diversification occurs from E12 followed by an expansion of ectodermal lineages from E13.5 onward. Consistent with the well-documented late emergence of amnion in most domestic animals compared to primates, the amnion cluster was not present in earlier samples but appeared later in gastrulation, from E12.5 onward. While it is possible that amnion may be involved in cell patterning in pig, as in primates, a role in A-P patterning is unlikely, as this occurs prior to amnion formation[24,25].

## Similarities between pig, human, monkey and mouse embryos

To gain insights into conserved and divergent features of non-rodent and rodent mammals we compared the transcriptomes of peri-gastrulation stage mouse[2] and *Cynomologous* monkey[6] embryos with pigs using high-confidence one-to-one orthologues (see methods). Projection of our pig dataset onto mouse[2], and subsequent transfer of mouse labels showed that the majority of cell-type annotations were well-matched between both species (Fig. 2a and Supplementary Fig. 2a), with large fractions of each cell type allocated being analogous to their mouse counterparts including cardiac mesoderm, extra-embryonic endoderm, spinal cord, primordial germ cells (PGCs), and epiblast 2 (Supplementary Fig. 2a). By contrast, later emerging neural cell types such as caudal and rostral neurectoderm, neural crest and fore/mid/hindbrain (FB/MB/HB) had fewer matches in our data consistent with neural tissues being more advanced in the mouse at the 10-somite stage compared to pig. In the case of extra-embryonic tissues,

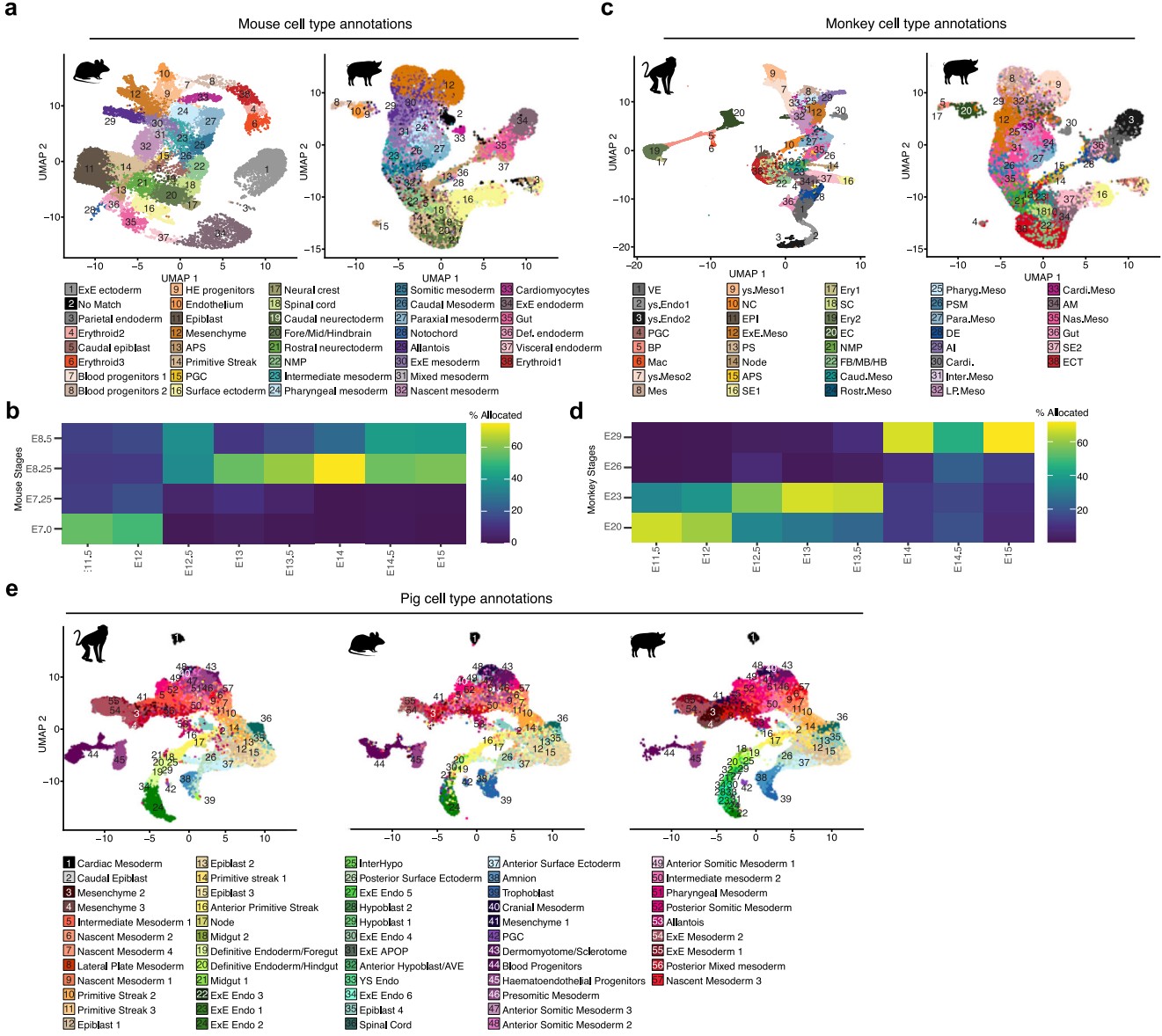

**Fig. 2 | Alignment of Pig, Mouse and Monkey datasets. a** UMAPs showing E6.5–8.5 mouse embryo cell types[2] and Pig E11.5 to E15 with mouse annotations after reciprocal PCA-based projection onto the mouse dataset. **b** Heat map showing the percentage of pig cells in each stage allocated to a particular mouse stage after label transfer. E Embryonic day. **c** UMAPs showing E20-29 monkey embryo cell types[6] and Pig E11.5 to E15 with mouse annotations after reciprocal PCA-based projection onto the monkey dataset. **d** Heat map showing the percentage of pig cells in each stage allocated to a particular monkey stage after label transfer. A percentage of 100 would indicate that all cells of a given cell type were predicted to be analogous to the cell identity in the queried organism. **e** UMAPs showing the aligned monkey, mouse and pig datasets with pig cell type and subtype annotations.

such as amnion and trophoblast, a large fraction of these tissues was allocated a surface ectoderm identity. Similarly, a large portion of extraembryonic mesoderm (ExM) was allocated as mesenchyme. Projections of pig stages onto mouse development show our time course aligns approximately between E7 to E8.5 (Fig. 2b). Projection mapping to the macaque dataset also showed a high degree of similarity of cell type annotations; in contrast to mouse-annotation mapping, this resulted in better agreement between predicted identities and our annotations of extra-embryonic mesodermal tissues (Fig. 2c; Supplementary Fig. 2b). Stage mapping also showed expected alignment between all our time points to monkey equivalents except E26 which had no clear match in our dataset (Fig. 2d). Given the large discrepancies in the methodologies and criteria used for annotating cell types in single-cell data[26] meaningful cross-species comparisons can often be challenging. To overcome these challenges, we used data projection/label transfer to apply our own cell type labels to each dataset for consistent annotation of equivalent cell types for further cross-species comparisons (Fig. 2e).

Hierarchal clustering of individual cell types (Supplementary Fig. 3) generally grouped cell types with lower correlation together, these included several extra-embryonic tissues (e.g., ExE Endo 1, 2, 4, 5 and Hypoblast 1 and 2) together corroborating known differences in the morphology and regulation of these tissues[3,25,27]. We noted that among cell-type specific marker genes, there was a large degree of overlap between monkeys, mice and pigs. This allowed us to determine sets of highly conserved cell type-specific markers: epiblast 1 (*POU5F1, SALL2, OTX2, PHC1, FST, CDH1* and *EPCAM*), PS 1 (*CDX1, HOXA1, SFRP2,* and *GBX2*), APS (*CHRD, FOXA2, GSC, CER1* and *EOMES*), node (*FOXA2, CHRD, SHH* and *LMX1A*) (Supplementary Fig. 4a–h), DE/Foregut (*SOX17, FOXA2, PRDM1, OTX2* and *BMP7*) and DE/Hindgut (*SOX17, FOXA2, TNNC1* and *ITGA6*). Notably, we also identified sets of genes that were strong cell type identifiers in monkey and pig cell

types, but not mice, for example, *UPP1, SFRP1, PRKAR2B, APOE* and *IRX2* in the epiblast, *CD9, GPC4* and *COX6B2* in the APS and *PTN, HIPK2* and *FGF8* demarcating node. We identified conserved and divergent markers for other less well-characterised cell types (Supplementary Data 3–6). These observations highlight the importance of investigating multiple representative animal models to identify conserved gene-regulatory networks relevant to cell fate decisions in mammals.

We then looked for transcriptional differences outside of cell-type specific gene programs, utilising ClusterProfiler to elucidate KEGG term enrichment among differentially expressed genes (DEGs). This revealed a considerable number of genes that were markedly upregulated in pig and monkey epiblasts compared to mice (Supplementary Fig. 5a). Further examination showed that a significant proportion of these DEGs were replicated across multiple cell-type comparisons. Notably, many of the identified genes were associated with distinct signalling pathways, including the Mitogen-Activated Protein Kinases (MAPK) and the Phosphatidylinositol 3-Kinases (PI3K)/Akt pathways, along with cell adhesion pathways such as those mediating focal adhesions and adherens junctions (Supplementary Fig. 5b). Given that these differentially expressed genes are part of pathways generally implicated in cell behaviour such as the regulation of cell growth, proliferation, differentiation, and morphogenesis, this may correspond to known differences in embryo size, cell cycle length and morphology between these species.

We next used our dataset to better understand the process of human gastrulation, currently informed by a single gastrula-stage CS7 (E17-19) human embryo[7]. Cross-species reference mapping of the CS7 human embryo onto our own dataset as well as that of mouse and monkey revealed heterochronicity between cell differentiation dynamics across species (Fig. 3a, b; Supplementary Fig. 6a–e). Intriguingly, despite the relatively immature stage of the human embryo, the mapping of mesodermal cell types onto their porcine counterparts

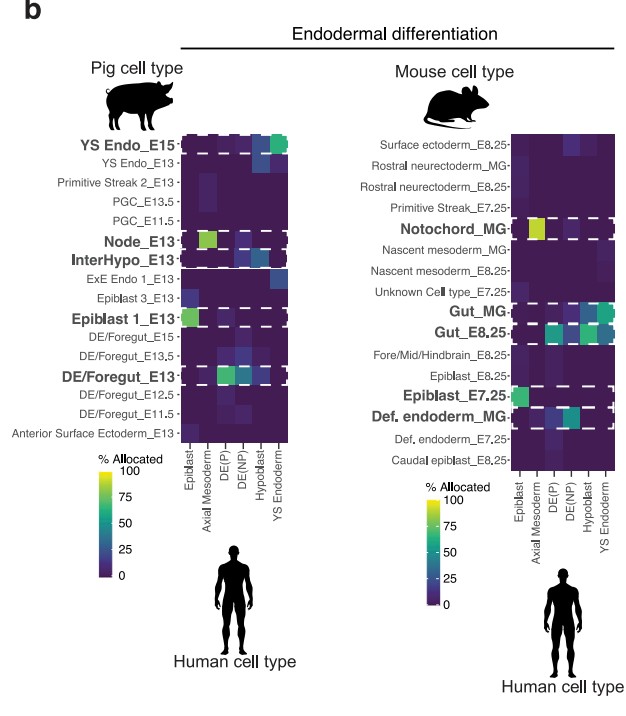

**Fig. 3 | Heterochrony across Pig, Mouse and Monkey differentiation. a** Heat maps showing the percentage of human mesodermal cells[7] allocated to a pig or mouse[2] cell identity after label transfer. **b** As with a, except with endodermal cell

types. 100% would indicate that all cells of a given cell type were predicted to be analogous to the cell identity in the queried organism. E Embryonic day.

revealed a considerable degree of alignment with E15 extra-embryonic mesoderm. This alignment potentially suggests that human extra-embryonic mesoderm not only envelops the embryo more extensively but also exhibits accelerated maturation when compared with other cell types, such as the epiblast and nascent mesoderm clusters. The latter two were found to correspond more closely with their E13 porcine equivalents, which more closely mirror the morphology of a CS7/8 human embryo. A congruous trend was observed comparing the human embryo to mice[2], as the human yolk sac mesoderm aligned closely to E8.5 mesenchyme (Fig. 3a). Comparisons of endodermal cell types also showed asynchronous development of yolk sac endoderm, like ExE mesoderm, human yolk sac endoderm had a higher mapping frequency to pig E15 yolk sac endoderm and E8.5 ExE endoderm (Fig. 3b). By contrast, nearly all the cell types investigated showed that CS7 human cell types matched CS9 in non-human primates (Supplementary Fig. 6d, e). While these results might reflect a discrepancy in embryonic staging, they also suggest little asynchrony between human and *Cynomolgus* monkey embryos. We also noted that annotations of ectodermal tissues such as amnion and surface ectoderm appeared to differ between human and monkey, while the pig annotations of these tissues aligned more closely to human (Supplementary Fig. 6a–c). This

suggested there is a need to better define the transcriptional profiles of these tissues for further comparisons and that gross morphology alone does not always indicate transcriptional equivalency. These findings suggest that although the programs guiding cell-type specific differentiation are remarkably conserved, the dynamics of differentiation exhibit notable variations across species.

## Pig mesoderm and endoderm progenitors are transcriptionally distinct

Given that our results suggested conservation in cell-type specific programs, we reasoned that the core mechanisms of differentiation would be conserved between mice and large mammals. However, one area of controversy is that of endodermal differentiation. Indeed, it has recently been suggested that proliferative, bi-fated "mesendodermal" progenitors are not found in the mouse embryo[16,17,28]. However, this idea has gained less traction in large mammalian embryology as early in vitro evidence from hESC differentiation studies has suggested that bipotent progenitors may exist[18–21]. Given the high numbers of cells of early gastrulation, our dataset allowed us to dissect the events of this period at high resolution. We analysed epiblast, PS, APS/node, nascent mesoderm, and DE clusters (Fig. 4a–d). Sub-clustering of mesoderm

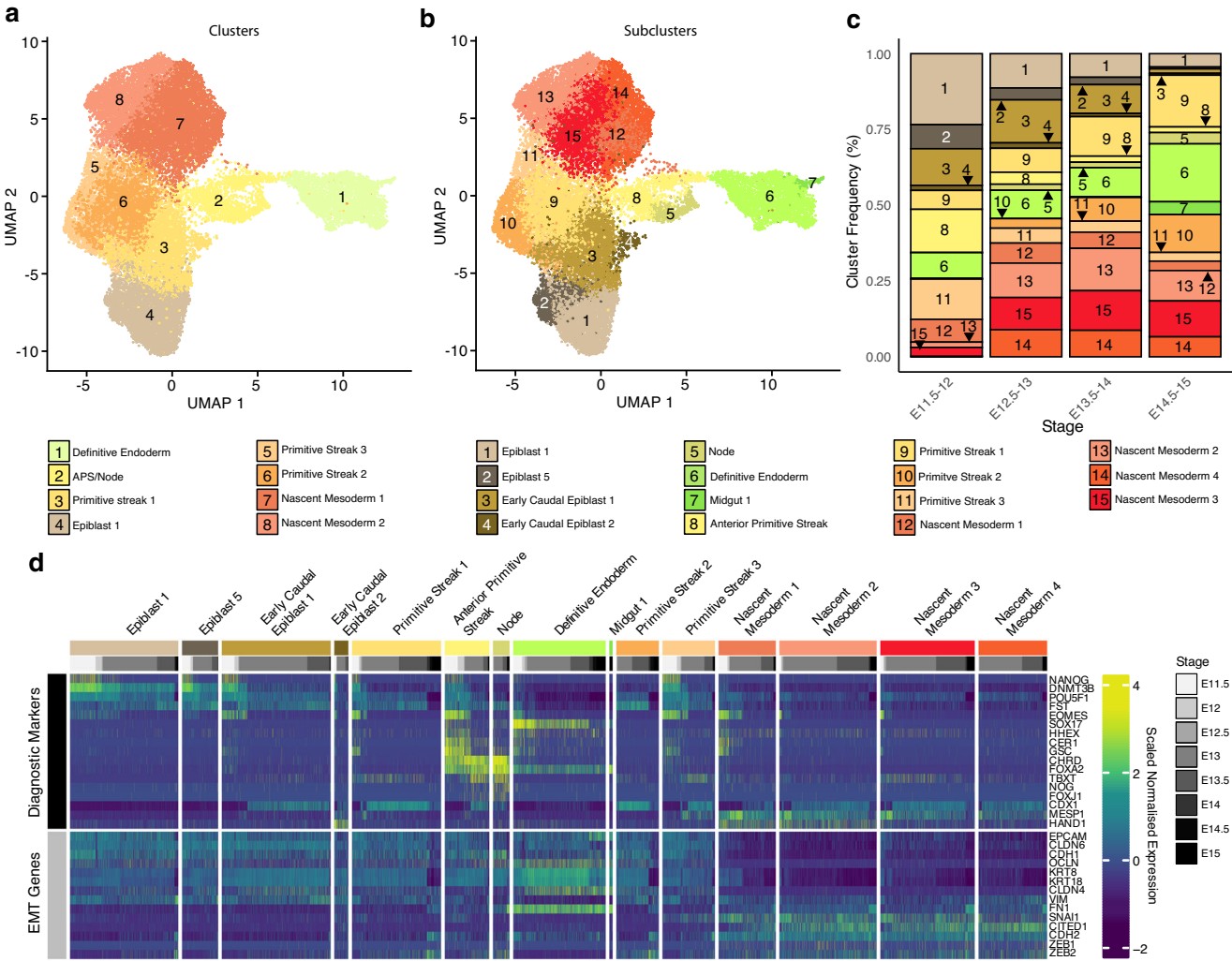

**Fig. 4 | Endodermal progenitors do not undergo classical EMT. a** UMAP plot showing epiblast, PS, APS/node, nascent mesoderm and DE clusters (24,874 cells; 23 biologically independent samples; E11.5-E15) coloured by global cell-type annotation and developmental time points. **b** As with a, coloured by cell subtypes. **c** Stacked bar graphs showing the frequency of each subcluster within the subset shown in **a** & **b** at specific time points in development. E Embryonic day. **d** Heat map illustrating the scaled expression of genes within individual cells. Expression of selected markers was used to identify cell subclusters as well as epithelial and mesenchymal marker genes.

and endoderm-fated cells identified 14 subpopulations: four nascent mesoderm, three PS, APS, node, two epiblast, two early caudal epiblast, DE, and midgut (Fig. 4b). Nascent mesodermal cells expressing *MESP1* increased in number between E11.5 to E14. Epiblast, early caudal epiblast and APS clusters decreased throughout time points (Fig. 4c). High-*SOX17* expressing DE cells were present from the earliest time point (E11.5) and throughout the time course (Fig. 4c, d; Supplementary Fig. 7). In contrast, node cells predominantly emerged one day later (E12.5). This finding is in line with previous observations where *FOXA2*, *TBXT*, and *GSC*-positive node cells can be identified from E12-E13 pigs[29,30], demarcating the start of secondary gastrulation in the pig. Early caudal epiblast could be distinguished by expression of *CDX1* and increased *EOMES* expression compared to epiblast 1 and 5. PS clusters 1 and 3 maintained expression of *DNMT3B* and had markedly higher expression of *TBXT* than caudal epiblast clusters. The PS2 cluster, by contrast, was not present until E12.5 onwards and showed reduced pluripotency expression and increased *CDX1* expression suggesting this may be later PS forming from the late caudal epiblast population. Nascent mesoderm was identifiable by decreased *DNMT3B* and *FST* expression and increased *MESP1*. We observed that most cells within the APS expressed both *FOXA2* and *CHRD*. Notably, however, the APS cluster manifested certain heterogeneity. A significant proportion of cells exhibited elevated expression of *POU5F1, NANOG, EOMES*, and *CER1*. Conversely, a subset of cells displayed lower levels of these markers, but higher expression of *TBXT*. These observations are consistent with the idea that distinct populations of the APS may give rise to the DE and node.

We next analysed the expression of epithelial markers and genes related to EMT (Fig. 4d; Supplementary Fig. 7). Tight junction markers *OCLN, CLDN6*, and *CLDN4*, along with the intermediate filament protein-encoding genes *KRT8* and *KRT18*, displayed low expression within nascent mesodermal clusters. Except for *CLDN6*, these markers also exhibited higher expression in the DE cluster. As with other epithelial markers, *CDH1* and cell-cell adhesion-associated *EPCAM* also showed reduced expression within nascent mesoderm compared to other clusters. In contrast, the expression of the mesenchymal transition regulator *SNAI1* and post-EMT marker *CITED1* showed elevated expression within the nascent mesoderm. Unexpectedly, the intermediate filament and mesenchyme marker *VIM* was expressed in most epiblast clusters but reduced in the APS, the node and DE. In addition, *CDH2* and *FN1*, frequently associated with EMT, were highly expressed within a portion of APS cells and the DE. Likewise, *ZEB2*, a transcriptional repressor of *CDH1*, was expressed in both nascent mesoderm clusters and the node.

Together these data suggest that cells of the APS, node, and DE, retain a robust epithelial identity throughout their differentiation despite upregulating a small number of genes associated with increased cell motility. This epithelial-to-epithelial transition has been previously described during the formation of the amnion in the mouse[31]. In contrast, PS-derived nascent mesoderm undergoes a divergent process resembling the "classic" model of EMT. Therefore, the processes by which epiblast cells transition from a columnar to a simple epithelium (in the case of DE), or toward a mesenchyme/mesenchyme-like state, as is the case for nascent mesoderm and notochordal process respectively, appear to be highly nuanced and tissue-specific. This observation is especially pertinent for the DE and nascent mesoderm, as we found no evidence supporting a common mechanism of cell ingression, in line with findings in mice[16,17,32].

## Exploring early somitogenesis in pig embryos

To explore the derivatives of the nascent mesoderm cells we subclustered nascent mesoderm, pre-somitic and somitic mesodermal cell types (Supplementary Fig. 8). This facilitated the identification of seven subtypes: 3 anterior somitic mesoderm clusters, cranial mesoderm, dermomyotome/sclerotome, posterior somitic mesoderm and pre-somitic mesoderm. We observed several genes with similar dynamics in pigs, as reported in mice and in human in vitro models. For example, *TBX6*, is highly expressed in pre-somitic mesoderm and posterior somites, but less so in more mature somitic cell types[33-35]. In contrast, *MYF5* and *MYOD1*, were expressed at later time points. Additionally, *FOXC2* was lowly expressed in all somitic subtypes. Generally, we observed the first mature somite subclusters emerge from day 14 onward while presomitic mesoderm clusters were present throughout, consistent with patterns described in many other species. We have also noted several markers of pig somitogenesis (Supplementary Fig. 8d).

## Spatiotemporal mapping of mesoderm and endoderm in pig

To elucidate the potential causal mechanisms underlying the formation of mesoderm, endoderm and node we applied reversed graph embedding and pseudo-temporal ordering to E11.5 to E13 epiblast, PS, APS/node, nascent mesoderm, and DE subclusters using Monocle3[36] (Fig. 5a–e). Epiblast and early caudal epiblast cells were positioned at the beginning of the trajectory, preceding the first bifurcation towards either PS/mesoderm or APS fates. Notably, few cells within the early caudal epiblast cluster appeared to be between PS and APS fates (Fig. 5a, b) suggesting the early caudal epiblast represents the last cells fated toward both mesoderm and endoderm. Comparisons of mesodermal and endodermal trajectories confirmed that in contrast to endoderm progenitors, mesodermal progenitors rapidly loose their epithelial characteristics and undergo "classical EMT" (Supplementary Fig. 9) as evidenced by elevated expression of *SNAI1* and *CITED1*. Trajectory analysis showed a secondary fate decision in the form of a bifurcation within the APS toward DE or node fates (Fig. 5a, b). Consistent with previous findings, *NANOG* expression was elevated in epiblast, early caudal epiblast, PS, and APS clusters but decreased sharply in node-fated cells and nascent mesoderm, in contrast to DE (Fig. 5c). We observed little difference in *FOXA2* expression between endoderm and node-fated cells, whereas *TBXT* expression was far more pronounced in node-fated cells of the APS.

Considering the absence of bi-fated mesendodermal progenitors outside the early caudal epiblast and the observed co-expression of *TBXT* and *FOXA2* in the APS/node, we investigated whether the APS fulfils the criteria for mesendoderm. Historically, the node/notochord has been categorized as a mesodermal tissue, as such, earlier descriptions of "mesendoderm" referred to the progenitors of the prechordal plate, notochord, pharyngeal and head endoderm[37-39]. Despite this classification, node cells initially migrate into the hypoblast layer as the notochordal process[40] and tend to cluster near the endoderm in low-dimensional space. Given the limitations of UMAP in inferring transcriptional similarity from spatial proximity, we employed module scoring with significant markers of mesoderm, endoderm, and epiblast to assess tissue similarity (Fig. 5d). While node cells exhibited a significantly lower endodermal score compared to endoderm itself it was higher than both epiblast and nascent mesoderm clusters. By contrast, the mesodermal score differences between DE and node clusters were not significantly different, suggesting the node is more transcriptionally similar to DE than mesoderm. Module scoring also validated our previous observations that DE cells had a higher epiblast score compared to mesoderm or node, aligning with the expression of pluripotency-associated genes in DE-fated APS cells.

Differential expression analysis highlighted several key factors differentiating PS/mesodermal from APS-fated cells after divergence from an early caudal epiblast state (Fig. 5e). PS/Mesoderm-biased states exhibited upregulation of *WNT8A, WLS* (WNT Ligand Secretion Mediator), and epigenetic regulators *MSH6* and *EZH2*. Conversely, APS-fated cells showed increased expression of *NODAL*, TGF-beta superfamily and NODAL-related factor: *GDF3*, along with *TLL1*, a metalloproteinase involved in processing TGF-beta superfamily precursors.

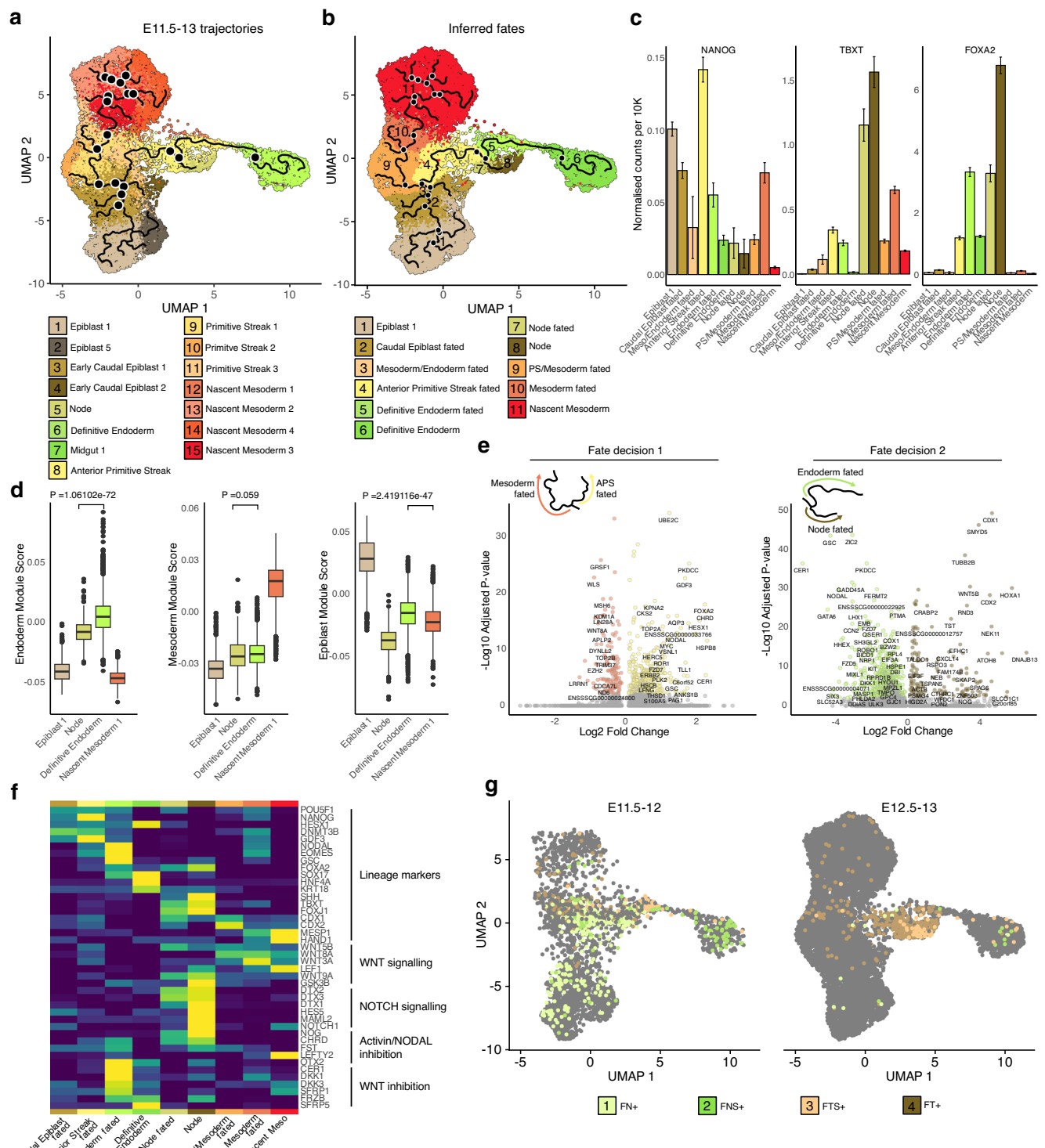

**Fig. 5 | Endoderm forms from Epiblast-like low TBXT progenitors. a** UMAP plot with reversed graph embedding trajectories projected on top using Monocle3. Black nodes mark trajectory branching points. $n = 16757$ cells across 11 biologically independent samples. **b** UMAP plot showing predicted cell fates inferred from Monocle3. **c** Bar graphs showing the NANOG, TBXT and FOXA2 expression in lineage-fated cells from a&b. Data are presented as mean values +/- SEM. **d** Box plots showing epiblast, nascent mesodermal and endodermal lineage scores in selected clusters from a. $n = 2340$, 263, 1536 and 1140 cells respectively across 11 biologically independent samples. Centre line represents median, minima and maxima hinges represent the 25th and 75th percentiles respectively. Whiskers extend from the quartiles to the last data point that is within 1.5 times the interquartile range. Points

beyond this range are shown and are considered outliers. *P*-values indicate the results of a two-sided Mann-Whitney U test. **e** Volcano plots showing differential expression between differently fated cells. Primitive streak (mesoderm fated) vs APS fates and Endodermal vs node-fated cells. Cut-off criteria for significant DEGs was a Log2 fold change ≥0.5 and an adjusted *p* value ≤ 0.01. **f** Heat map illustrating the scaled average expression of selected genes in each of the cell fates identified in **b. g** UMAP plots showing cells categorised by FOXA2, NANOG, TBXT and SOX17 expression at selected time points. F FOXA2, N NANOG, T TBXT, S SOX17 cells. Cells are coloured by their F/N/T/S category. APS Anterior primitive streak, E Embryonic day.

Differential expression analysis also provided insights into DE vs. Node fate decisions, with DE-fated cells continuing to display increased expression of *NODAL* as well as known endodermal regulators such as *GSC* and *CER1*. Node-fated cells showed increased expression of retinoic acid modulator *CRABP2* as well as upregulation of caudal factors such as *CDX1* and *CDX2*. Given the differential expression of several genes involved in signal regulation, we looked at the expression of genes involved in active WNT signalling, NOTCH signalling, Activin/NODAL inhibition and WNT inhibition (Fig. 5f). In line with many of our previous observations we observed increased expression of factors involved in WNT signalling in mesoderm and node fated cells albeit these tissues showed differential expression of specific WNTs. In contrast to mesoderm, however, node-fated cells showed high expression of Activin/NODAL inhibitors *NOG*, *CHRD* and *FST* and upregulated several genes involved in NOTCH signalling. DE-fated cells showed far greater expression of WNT inhibitors compared to their node-fated counterparts or mesoderm-fated cells.

While we did not observe high levels of *TBXT* and *FOXA2* outside of the APS and node clusters, we looked for rarer cell types that may co-express *TBXT* and *FOXA2* that exist in other cell clusters. As we have previously shown NANOG and SOX17 are exclusively expressed in the ED and hypoblast layer respectively in pig E10-E11.5 embryos[41], we reasoned that including the expression status of these markers in conjunction with *FOXA2* and *TBXT* could be used to ascertain the position of cells during fate commitment. Of note we identified, 287 *FOXA2 +, NANOG +, TBXT-, SOX17-* (FN+) cells, 98 *FOXA2 +, NANOG +, SOX17 +, TBXT-* (FNS+), 500 *FOXA2 +, TBXT +, NANOG-, SOX17-* (FT+) cells and 110 *FOXA2+, TBXT+ SOX17+* (FTS+) cells (Fig. 5g). We observed that of these groups FN+ cells were most abundant at E11.5-12, while FT+ cells were more abundant at E12.5-13 concurrent with the fate switching of the APS from DE to node that also occurs during this period. In line with the high levels of *TBXT* and *FOXA2* expression in node-fated cells, FT+ cells made up a large percentage of this group and were less abundant in the DE-fated group (Supplementary Fig. 10a, b). FN+ cells were more abundant in early caudal epiblast, APS and DE-fated cells (Supplementary Fig. 10a). We subsequently examined whether cells that co-express mesodermal and endodermal factors exhibit concurrent high-level expression of these factors, or if their expression patterns are mutually exclusive (Supplementary Fig. 10b). *TBXT* showed a positive correlation with *FOXA2* both in cells with detectable *SOX17* ($R = 0.33$, $p < 0.0001$) and without ($R = 0.48$, $p < 0.0001$). *FOXA2* also had a significant correlation with both *SOX17* ($R = 0.37$, $p < 0.0001$) and *EOMES* ($R = 0.16$, $p < 0.0001$) only in FN+ cells which did not have detectable *TBXT* expression. *EOMES* expression was negatively correlated with *TBXT* in cells with both *SOX17* positive ($R = -0.36$, $p < 0.001$) and *SOX17* negative cells ($R = -0.53$, $p < 0.001$). *CDX2* expression did not correlate with any other factor but *TBXT* which was limited to FT+ cells ($R = 0.14$, $p < 0.01$). Together these data suggest that cells co-expressing high levels of *TBXT* and *FOXA2* primarily contribute to the node/notochord. Given the APS origin of the node, endoderm-like transcriptional signature, and that APS/node cells like endoderm do not appear to undergo classical EMT this also suggests that FT+ cells are not mesendodermal at all. Likewise, it appears that most cells that contribute to DE have no detectable *TBXT* transcripts or low levels and correspondingly most mesoderm-fated cells are *FOXA2* negative. We also find these observations apply in a less binary fashion to mouse embryos, with the high *T* population being node/notochord fated and a low *T* population, endoderm fated (Supplementary Fig. 11).

### FOXA2 and TBXT cells in gastrulating pig embryos
To spatially position mesodermal and endodermal progenitors, we conducted whole-mount immuno-fluorescence imaging of E10.5-E11.5 porcine embryos for SOX2, FOXA2, and TBXT (Fig. 6; Supplementary Movies 1-3). At E10.5, TBXT-positive (T+) cells appear in the posterior epiblast, their numbers increased in E11.5 embryos. Many of these cells extend beyond the posterior ED boundary as ExM by the end of E11.5

(Fig. 6a–d). A group of FOXA2-positive TBXT-negative (F+) cells was detected anterior to the FOXA2+ and TBXT+ (FT+) cells at E11.5. A similar population of F+ cells in the epiblast layer was recently reported in mice[16,17]. While initially, this F+ population outnumbered FT+ cells by a factor of ~3, this gradually decreased to ~1.4 by the end of E11.5 (Fig. 6c, Supplementary Fig. 12). Lateral reconstructions of late E11.5 embryos showed F+ cells part-way delaminating from the epiblast into the hypoblast layer (Fig. 6d). In line with their transcriptional profiles, early epiblast-derived F+ cells contributing to the hypoblast/DE layer are NANOG +, in contrast to the T+ cells which are NANOG negative. By E12 NANOG+ cells are rarely detected (Fig. 6e). Importantly, we did not observe any FT+ cells intercalating the hypoblast. In line with their spatial positioning and the transcriptomic profiles (Figs. 4, 5, 6), these observations indicate that the F+ population is fated to DE while the FT+ cells represent node/notochord precursors. This is further supported by the increased number of double-positive TBXT/FOXA2 (FT+) cells demarcating the medial APS region by the end of E11.5.

### Extraembryonic signalling correlates with emergence of DE
Previous studies demonstrated the contribution of extra-embryonic endoderm to DE in mice[2,42–44]. To establish whether this is true in pig embryos we isolated DE, gut, hypoblast and extraembryonic endoderm (ExE) clusters for further analysis (Supplementary Figs. 13a–d, 14–16). Through marker expression we identified subclusters of the hypoblast including the posterior hypoblast and AVE as well as yolk-sac endoderm (Supplementary Figs. 13d, e, 14). Temporal dynamics and module scoring also revealed a population of cells that appear to be an intermediate (InterHypo) between hypoblast and DE in line with similar observations in mice[42] (Supplementary Fig. 13f–g). We also confirmed previous observations that suggested the hypoblast is the primary source of *NODAL* in pig embryos[45] (Supplementary Fig. 13h–i). However, *NODAL* signalling in the hypoblast is restricted to E11.5-E12, which coincides with the specification of the DE from epiblast cells. Moreover, the absence of NODAL at E12.5 also correlates with the displacement of the hypoblast with DE and the fate switching of the APS cells from endoderm to node/notochord. These findings are consistent with our observations that APS-fated cells upregulate *NODAL* expression before node-fated cells upregulate NODAL inhibitors. Indeed, recent reports have suggested that timed Activin/NODAL inhibition promotes notochord formation from DE-competent cells[46,47]. Lastly, the isolation of E14-E15 samples allowed for the identification of gut sub-populations that resemble their mouse counterparts[43] (Supplementary Figs. 15, 16).

### Organiser-like signalling patterns of porcine cell types
It has been suggested that in mice, the node, PS and hypoblast have functions analogous to the organizer in amphibians[48–50], yet it is unclear whether the signals leading to A-P patterning are relevant to other mammalian species. One such example is *Wnt3*, secreted by the posterior epiblast in response to *Bmp4*, which acts as the primary driver of mouse gastrulation[51]. Experiments in hESC have demonstrated that, like in mice, WNT3 is the only WNT orthologue to respond to BMP4 stimulation[52]. However, while mouse Wnt3 knock-out embryos fail to gastrulate[51], humans with a homozygous WNT3 nonsense mutation can complete gastrulation but develop tetra-amelia and urogenital defects[53]. Furthermore, the only demonstration of WNTs organiser function in large mammals comes from in vitro experiments using WNT3A[52]. Given the lack of understanding of the in vivo role of WNTs in large mammals, we investigated canonical WNT crosstalk in E11.5-E12 cell types using CellChat[54] (Supplementary Fig. 17a–c). In line with findings in mice and in keeping with a role in A-P patterning, many early cell types were highly receptive to WNT ligands via multiple FZD and LRP receptor combinations. The highest *WNT3* signals came from the PS and APS

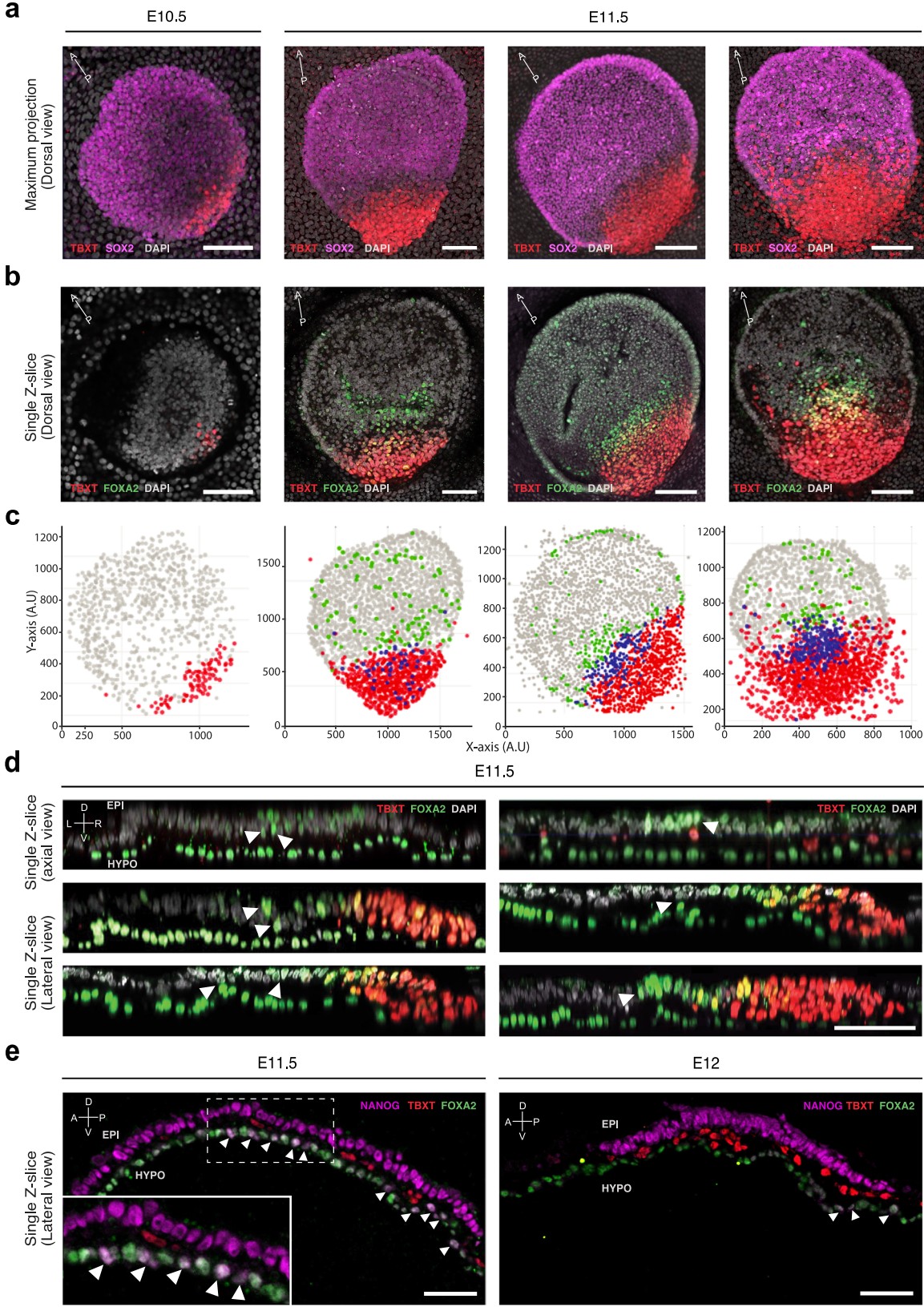

**Fig. 6 | FOXA2 and TBXT domains are spatially separated. a** Maximum intensity projection (Dorsal view) of E10.5 (*n* = 1) to E11.5 (*n* = 4) porcine embryos showing TBXT and SOX2 expression. E11.5 embryos are ordered left to right by age. E Embryonic day. **b** Single z-slice of the embryos shown in **a** showing FOXA2 and TBXT expression. Scale bar: 50 μm. **c** In Silico representations of embryos following 3D segmentation of embryos from **a** and **b**. **d** Axial and lateral reconstructed sections of embryos stained for FOXA2 and TBXT. Epiblast layer is oriented above the hypoblast/DE layer. White arrowheads indicate FOXA2 + TBXT- cells that are spatially separated from the TBXT domain. Scale bar: 50 μm. **e** Lateral sections of E11.5 (*n* = 1) and E12 (*n* = 1) embryos, showing expression of NANOG, FOXA2 and TBXT. Epiblast layer is oriented above the hypoblast/DE layer. White arrowheads indicate NANOG+ cells. Scale bar: 50 μm. Please refer to Supplementary Fig. 21 for a colour blind friendly version of the figure.

(Supplementary Fig. 17a, b). Conversely, *WNT3A* was expressed within anterior ED clusters, namely the epiblast and APS.

Cell-cell signalling analysis also showed that multiple cell types were predicted to be receptive to node-produced SHH (Supplementary Fig. 17c). Notably, the scarcity of node cells identified prior to E12.5, is consistent with the notion that the mammalian node/notochord is principally involved in secondary gastrulation[48]. In line with this hypothesis, node cells and node-fated cells not only secreted SHH, a dorsal tissue patterning ligand, but also expressed high levels of *NOG* and *CHRD*, factors primarily associated with axial extension and DV patterning[55–57].

### WNT/NODAL guides DE differentiation directly from Epiblast

WNT and NODAL signalling is critical for endoderm formation[52], however, the balance of these competing signals required for DE formation has not been fully established. To assess this, we used varying concentrations of Activin A (ActA) and a WNT agonist (CHIR99021, CHIR) to simulate specific microenvironments across the ED in 2D cultures of the pig EDSCs line (EDSCL4)[22] and hESC lines H9 and HNES1[58]. All cells were maintained undifferentiated in AFX medium (see methods). Upon differentiation, a dose-response effect to the addition of ActA was observed with the number of FOXA2 positive cells decreasing when combined with higher CHIR levels at both 24 and 48 hrs in pEDSCs. This effect was not observed in H9, where only a modest increase in FOXA2 protein was determined after 48 hrs. In addition, pEDSCs showed higher SOX17 expression by 48 hrs when exposed to increasing levels of ActA (20–100 ng/ml) in the presence of low CHIR levels but showed a stunting of SOX17 expression in response to higher CHIR. A similar trend is seen in H9, but H9 also showed an increased sensitivity to ActA, with peak SOX17 expression at 20 ng/ml and a comparatively reduced levels when treated with 100 ng/ml. In pEDSC endogenous WNT inhibition using XAV939 resulted in very low SOX17 expression, indicating a moderate level of WNT is required for endoderm differentiation (Fig. 7a, b, Supplementary Fig. 18).

In line with recent reports[59], TBXT expression required exogenous Activin/NODAL, in addition to WNT, suggesting a role for the hypoblast in streak formation. In both species, higher levels of CHIR increased TBXT expression at 24 hrs, which was largely extinguished by 48 hr, consistent with the TBXT expression profile in the embryo. Co-expression between SOX17 and TBXT or SNAI1 was rarely observed at 24–48 hrs and showed very low Pearson's correlation scores (Supplementary Fig. 19a–d). TBXT/SOX17 co-expression was not observed at 8 hrs (Supplementary Fig. 19e, f).

Previous studies showed that there are differences between using WNT3A and CHIR, both in cellular response and the efficacy of in vitro differentiation of hESC towards DE[60,61]. We treated pig EDSC with WNT3A and found minimal response to WNT3A alone, however robust expression of FOXA2 and SOX17 was determined when ActA was added (Supplementary Fig. 20). In contrast, H9 hESC exposed to WNT3A alone upregulate FOXA2, however, like in pEDSCs, SOX17 was only upregulated when ActA + WNT were added. In hESC, expression of both FOXA2 and SOX17 was significantly enhanced by WNT stimulation compared to CHIR stimulation, a response not seen in pEDSC. These experiments show that DE can be induced more efficiently with WNT3A, as previously demonstrated[61], however, the sensitivity to this inducer differs between pig and human lines.

Based on these results we propose a model of DE formation in the pig embryo (Fig. 7c) whereby between E10.5 and E11, the first TBXT+ cells emerge at the posterior-most point of the ED opposing the anterior-most hypoblast/AVE-produced WNT/NODAL inhibitors (i.e LEFTY2 and DKK1)(Supplementary Fig. 13d, e)[45]. Driven by a combination of posterior epiblast-secreted WNT and hypoblast-derived NODAL signalling, these cells initiate EMT, thus constituting the nascent mesoderm. Concomitantly, in cells anterior to the TBXT ED domain a combinatorial effect of WNT signal inhibition and increased NODAL activity induces EOMES. EOMES, which is capable of inhibiting TBXT gene regulatory activity[62], alongside NODAL, promotes *FOXA2* expression. The most anterior FOXA2+ cells secrete NODAL inhibitor CER1, delaminate, and intercalate into the hypoblast to form and expand the DE. This results in the termination of NODAL signalling in the newly formed DE and WNT signal inhibition in the APS. The remaining FT+ NODAL-primed cells, still driven by WNT signalling, begin to form the node from approximately E12 onward.

## Discussion

We present a scRNAseq atlas of pig gastrulation and early organogenesis that represents a comprehensive resource for exploring the molecular mechanisms governing cell-fate determination during a crucial juncture of development. We harnessed this resource to investigate the major temporal signalling and differentiation events, which direct primary gastrulation in bilaminar disc embryos. Our findings underscore the nuanced heterochrony in embryonic development across mammals, evidenced by asynchrony in cell type maturation[63]. Further, these heterochronic differences combined with comparisons between pigs, monkeys and mice reinforce the idea that extraembryonic tissues may be less conserved than their embryonic counterparts[3,25,27]. Through cross-species mapping and comparative transcriptomics, we also elucidated distinctive gene expression patterns associated with cell-cell adhesion and signalling pathways in pig and monkey epiblasts, compared to their mouse counterparts. Such observations may have implications for understanding species-specific aspects of cell differentiation, growth, and morphological features. Our findings also highlight the ongoing need for comprehensive and well-annotated single-cell atlases of mammals to better characterise in vitro embryo models. Frequently, models such as mice have been used to "stage" in vitro models of other species, such as humans. However, given that the stage of differentiation of a given cell-type may differ between species despite morphological/structural similarities, this approach may not always be appropriate.

Despite the observed species differences, we were able to identify highly conserved early cell-type specific programs between mice, primates and pigs. These findings were exemplified by our investigations into the segregation of the endoderm, mesoderm and node, the precursors of which display organizer-like patterns of gene expression. Without the reduced spatial constraints and slower development of the pig embryo compared to mice, we can show a distinct temporal and spatial pattern of lineage specification from ED cells. Contrary to earlier proposed models in hESC, which suggest a bipotent mesoderm-endoderm progenitor undergoes EMT prior to lineage specialisation, we find little evidence of this.

Instead, we observe that classical EMT predominantly occurs in PS cells transitioning to nascent mesoderm. Importantly, this transition occurs following the segregation of the PS and APS from an early caudal epiblast population. Given the expression of pluripotency and epithelial markers, this suggests that the last cells capable of producing both mesodermal and endodermal populations are situated in the epiblast layer. APS and subsequently node and DE populations maintain their epithelial characteristics during their ingression in a mechanism that resembles an epithelial-to-epithelial transition. These observations in the pig embryo are supported by recently proposed murine models[16,17,28]. Given that the early caudal epiblast population itself can be characterised by the expression of factors that are not associated with ectoderm-fated cells (e.g., *EOMES*, *CDX2*) this raises questions about the extent to which this epiblast population can be considered bipotent or merely predisposed toward mesendodermal fates.

We also looked for the co-expression of mesodermal and endodermal genes and while most mesodermal and endodermal genes showed mutually exclusive expression profiles, genes such as

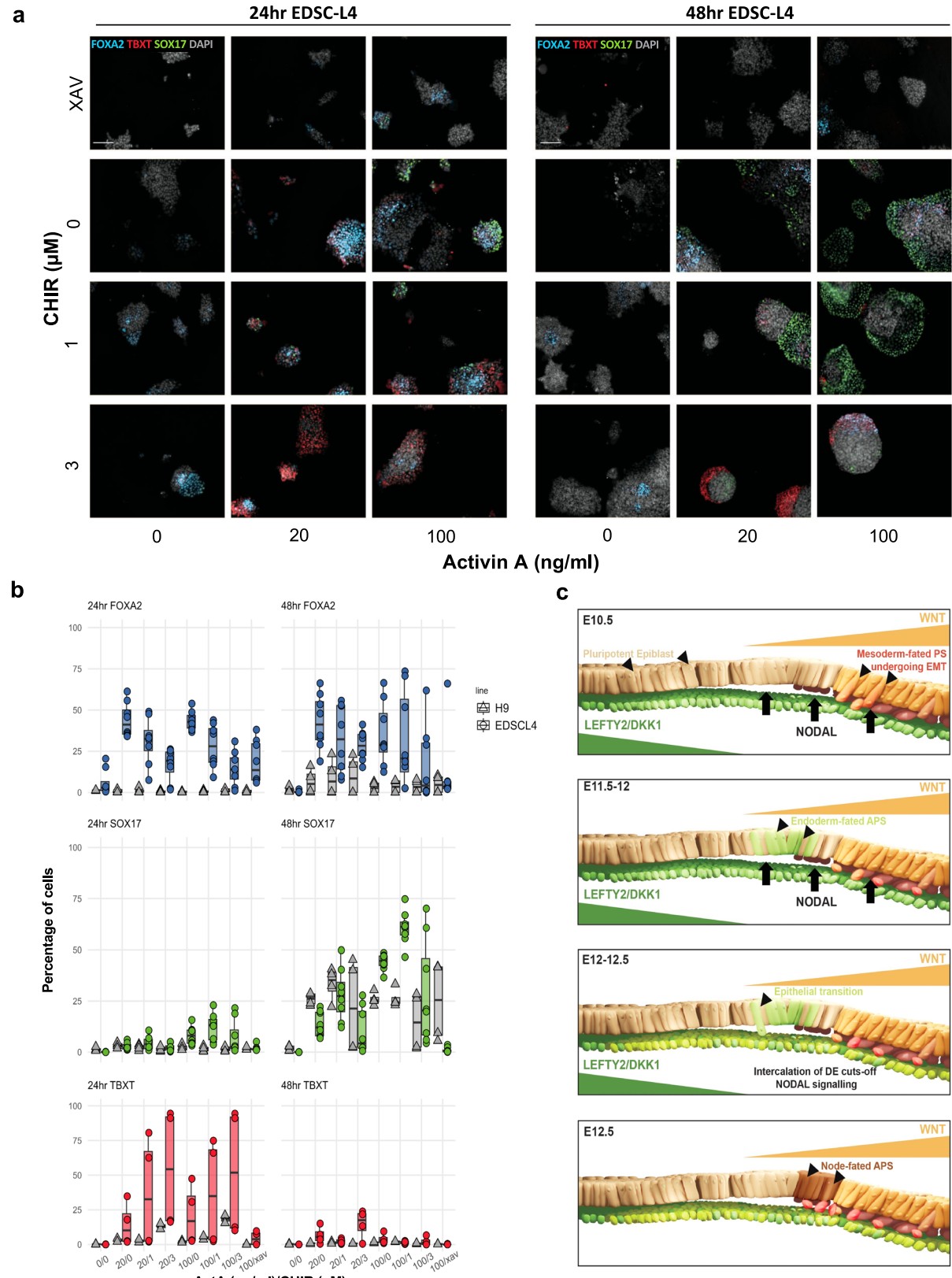

**Fig. 7 | Effect of Activin A and WNT signaling in pig EpiSC and hESC.**
**a** Representative images depicting differentiation conditions for EDSCL4. Images were captured using an Operetta CLS high-throughput microplate imager. Scale bar: 200 μm. **b** Box plots summarizing 2D differentiation experiments. Data is normalised to well background signal. *n* = 3 independent experiments. Centre line represents median, minima and maxima hinges represent the 25th and 75th percentiles, respectively. **c** Proposed model of epiblast-DE differentiation in pig embryos. Please refer to Supplementary Fig. 22 for a colour-blind-friendly version of the figure.

*TBXT* and *FOXA2* are both highly expressed in node progenitors as well as within the node/notochord. Earlier work describes this population, which gives rise to a portion of anterior endoderm, as mesendodermal[37–39]. This definition is therefore predicated on the notochord itself being a mesodermal tissue. Because of this, we considered whether node/notochord was of mesodermal origin in the pig embryo. Notably, lineage scoring of mesodermal, endodermal and node populations revealed that node/notochord is transcriptionally closer to endoderm than mesoderm. However, despite this result, the unique signalling profiles, transcriptional signatures, and nuanced differences in EMT gene expression within this population suggest the node may not fit classical definitions of either mesoderm or endoderm.

Our findings extend the understanding of mesodermal, endodermal and node progenitor organiser-like signalling patterns. We show that node progenitors in the APS and later arising node do not express A-P patterning genes (*NODAL, CER1, LEFTY2* and *DKK1*), but instead express genes related to axial extension. These findings are in contrast with chick models where the node appears to have a clear role in A-P patterning[48]. Combined with both embryo whole mount IF and in vitro experiments we showed that this sequence is determined by a balance of WNT and Activin/NODAL signalling gradients along the A-P and D-V axis, respectively, and equivalent findings were recapitulated in vitro with hESC, suggesting that these findings are representative of human embryos. Altogether, these findings provide evidence that classical EMT occurs as primitive streak cells lose their pseudostratified epithelial organisation as they transition to nascent mesoderm. This classical EMT occurs after the segregation of the primitive streak and anterior primitive streak from a "caudal epiblast" population. This suggests that the last cells capable of producing both mesodermal and endodermal populations are situated in the epiblast layer. Given that the APS and subsequently node and DE populations maintain their epithelial characteristics during their ingression they likely internalise via mechanisms independent of those that govern mesodermal ingression.

While we have identified several conserved and divergent features of mammalian gastrulation it is important to recognise the limitations inherent in such analyses. Firstly, differences in the quality of reference genomes/transcriptomes and annotations, which form the basis for quantifying transcriptional programs vary greatly between species, potentially leading to discrepancies in data quantification and interpretation. Additionally, assumptions about the orthology of genes and indeed the exclusion of genes in which orthology cannot be assumed will undoubtedly mean many factors that may be critical in cell-fate determination in a particular species will be overlooked. Lastly, given the logistics of collecting and sequencing samples of a given species we have made our comparisons with publicly available datasets of which the methodologies surrounding embryo collection, handling, dissection, and single-cell isolation may vary and therefore batch effects may mask biological variation. Therefore, while our findings provide valuable insights into conserved and divergent aspects of development, they also underscore the need for caution in extrapolating results across different species. By addressing these conceptual challenges, we can better harness the potential of such cross-species analyses to reveal fundamental principles of development.

We anticipate that this resource in combination with other recent works[3] and techniques, such as spatial transcriptomics of early embryos[64], will inform more detailed analyses of species differences and will shed light on the molecular events that underly the phenotypic differences in mammalian embryos. Given their use in agriculture pig embryos represent a highly accessible model for functional investigations relevant to human development. As with comparable datasets in mice[2], we expect that this dataset will serve as a wild-type reference for comparisons against mutant embryos. Such

investigations will serve as a foundation on which to develop more robust in vitro differentiation protocols of pluripotent cells, improved methodologies to produce interspecies chimaeras and to study the development of organs used for xenotransplantation[9–11].

## Methods

### Ethical statement
All procedures involving animals were approved by the Animal Welfare and Ethics Review Committee (Nbr. 99) of the School of Biosciences, The University of Nottingham. The research conducted adhered to the Home Office Code of Practice guidelines for the Housing and Care of Animals used in Scientific Procedures.

### Embryo collection
Animals used in this study were monitored twice daily and sacrificed by electrical stunning followed by exsanguination (Schedule 1). Embryos were collected from crossbred Large White and Landrace sows (2–3 years old) between days 10 to 16 post artificial insemination. Each uterine horn was separated into an upper and lower half before fresh PBS + 5% BSA was used to flush out porcine embryos. Uterine horns were then bisected and searched by hand for any further embryos. Embryos were stored in warm PBS + 5% BSA during. Embryos were either fixed in PFA at 4 °C overnight for IHC or taken for 10x single-cell RNA sequencing. To ensure a representative sampling of cell types and to mitigate the overrepresentation of cells from extraembryonic tissues, the embryos were carefully dissected to remove most of these tissues prior to dissociation. Embryo dissections were performed using forceps, and extraembryonic membranes were carefully dissected, avoiding disrupting ED derivatives.

### IF and imaging of whole-mount embryos
Embryos were permeabilized and blocked at the same time for 2 hr in a solution containing 10% donkey serum and 5% BSA with 1% Triton-X (PB buffer) at RT. Samples were incubated with primary antibody O/N at 4 °C in PB buffer. Secondary antibodies were incubated with sample for 2 hr at RT in PB buffer. Washes were performed after primary and secondary antibody incubations 4x15min in PBS with 0.2% Triton-X. Samples were mounted in either VECTASHIELD or Fluoroshield with or without DAPI. If the mounting media did not contain DAPI, it was added at the secondary antibody incubation stage. Antibodies used here are listed in Supplementary Table 1. Imaging of embryo sectioned and whole mount samples was performed with a confocal Zeiss LSM 900 with Airyscan, samples imaged as z-stacks were done so with a Z resolution of 0.32 μm.

### Segmentation and quantification of embryos
Z stack images were segmented using a StarDist/TrackMate pipeline within Fiji. StarDist allows for 2D segmentation, and TrackMate is used to build up each cell in 3D from the totality of the 2D segmentation data. 3D data for each cell was tabulated and the data for each embryo was exported as a.CSV file and further analysed in R using custom code. Thresholding was achieved via Otsu's method within fiji. Analysis in R allowed for the extraction of location and protein expression information for each cell, which was then used to recapitulate the embryos in silico, as well as directly quantify the number of cells expressing a given protein. Lateral re-slicing performed within Fiji.

### Culturing, imaging and quantification of 2D experiments
Both human and pig cells were cultured in AFX medium: N2B27 supplemented with FGF2 (20 ng/ml) + Activin A (12.5 ng/ml) + XAV939 (2 μM)[22]. For experimental procedures, cells were seeded at a range of seeding densities (750–1200 cells/well) into CytoOne TC treated 96 well plates coated with Laminin-511-E8 fragment with additional ROCKi for 48hrs. Cells were differentiated in N2B27 supplemented with CHIR, Activin A, XAV and WNT3A

depending on the condition. Cells were then fixed with 4% PFA and immunostained as detailed above. Images were captured using an Operetta CLS high-throughput microplate imager for the CHIR experiment or a CellDiscoverer 7 confocal plate reader for the WNT3A experiment. Images were segmented using Harmony (v4.1) from phenoLOGIC or StarDist2D within Fiji, respectively. Thresholding, cell expression identity and plotting was performed with a custom script in R. Cell lines used were EDSCL4 (porcine), H9 and HNES1 (human) (Supplementary Table 1).

**Single-cell transcriptomic analysis**
**Preparation of scRNA-seq library and sequencing.** Single-cell libraries were constructed using Single Cell 3 Library & Gel Bead Kit v3 according to the manufacturer's protocol (10X Genomics). Briefly embryos taken for scRNAseq were either frozen and stored at −20C or sequenced fresh. Given the large size of ExE tissues, to ensure that embryonic cells were well represented, we manually dissected most ExE structures prior to dissociation and sequencing, preserving the hypoblast. Embryos were dissociated into single cells via incubation with TrypLE, for 7 mins at 37 °C. Embryos were further dissociated via pipetting and then washed with a solution of DMEM/F12 0.04%BSA to quench TrypLE activity. Cells were strained though a Flowmi cell strainer into a 15 ml falcon. Once dissociated, cells from pools of same stage embryo were counted using a haemocytometer. Approximately 4000–14000 cells per lane of each 10x chip were transferred. The chip was then loaded into a Chromium Controller for cell lysis, cDNA synthesis and barcode labelling. cDNA libraries were assessed using an Agilent 2100 Bioanalyzer (Agilent Technologies). Finally, the libraries underwent 150 bp paired-end sequencing using the NovaSeq 6000 platform.

**scRNA-seq data preprocessing.** Raw fastq files were processed using cellranger-6.1.2 software with default mapping arguments. Reads were mapped to the Sus scrofa Sscrofa11.1 (GCA_000003025.6) genome. Subsequently, the CellRanger 'aggr' command was used to normalize the sequencing depth of different samples. For samples where the number of sequenced cells differed greatly from the counted number of cells prior to loading the 'force cells' command was used, and the estimated number of loaded cells was given.

**Filtering, dimensionality reduction and clustering.** The filtered expression matrix with cell barcodes and gene names was loaded with the 'Read10X' function of the Seurat (v.4.0.0) R package[65]. Following this, Seurat objects from each sample (23 samples) were independently created and processed according to standard Seurat protocols. Initially, single cells with a number of detected genes (nFeature_RNA) above 1750 were retained to exclude low-quality cells. Subsequently, doublet or multiplet cells were identified with the DoubletFinder R package[66] and excluded. After normalization of the Seurat object, we selected the 3000 most variably expressed genes using the 'FindVariableFeatures' command. These features were used to calculate the first 100 principal components. Given the heterogeneity of cell compositions of early and late embryos, we did not exclude cells based on the percentage of their transcriptomes that were made up of mitochondrial genes. Instead, following clustering, we looked for non-discreet clusters that had significantly higher expression of mitochondrial genes and low 'nFeature_RNA'. Two clusters of cells were excluded on this basis, we also noted that these cells showed expression of markers of multiple cell-types suggesting that these cells were clustering based on a shared apoptotic identity. We then used the 'FindIntegrationAnchors' and 'IntegrateData' functions of Seurat to exclude individual heterogeneities between samples. Data integration was done using the reciprocal pca (rpca) method and the first 30 dimensions of each object were used. To construct the main UMAP plot of 91,232 cells in Fig. 1c, we used the first 25 principal components

for calculating UMAP 1 and 2, setting the seed at 42, using a minimum distance of 0.5 and the 50 nearest neighbours (n.neighbors). The other parameters were kept as the defaults for UMAP generation. For clustering of the same cells, the 'k.parameter' of 20 and 'n.trees parameter' of 50 were the default settings during the neighbour-finding process; 25 dimensions were selected via the ElbowPlot method for neighbour finding, and clustering. A resolution of 1.2 was used to identify the 36 major cell-types. For sub-clustering, cell-types of interest were subsetted, objects were re-scaled and then the same numbers of principle components were calculated. The same parameters were used for neighbour finding, however, a resolution of 0.5 was used to cluster. For UMAP creation, again the ElbowPlot method was used to select the number of dimensions however the 30 nearest neighbours were used along with a minimum distance of 0.4. We calculated the DEGs of each cell cluster with RNA assay using the 'FindAllMarkers' function in Seurat. Heatmaps were plotted based on the most highly expressed genes (according to fold change) which had an adjusted p-value less than 0.05.

**Pseudotime.** The 'monocle3' R package[67] was used to calculate the developmental pseudotime of E11.5-E13 epiblast, PS, nascent mesoderm, APS, DE and node subclusters. The Seurat object was converted to a monocle3/ CellDataSet class by the 'as.cell_data_set' command of the SeuratWrappers R package. Cells were then processed according to the developer vignettes and fate information was transferred back to the Seurat object for further analysis.

**Gene expression-based categorisation of cells.** Cells were categorised as having 'positive' expression of *NANOG, FOXA2, TBXT* or *SOX17* cells if they had a scaled expression value greater than 0.1.

**Transcription factor regulon analysis (SCENIC).** Gene regulatory networks and regulons were elucidated using the command-line interface (CLI) of the pySCENIC pipeline, including tools such as 'arboreto_with_multiprocessing.py', 'ctx', and 'aucell'[68]. Input data consisted of raw gene counts, pre-processed to consider cells detecting between 1,750 to 7,500 genes. Cells bearing over 10% mitochondrial reads, along with genes identified in less than three cells, were excluded from the analysis. For transcription factor binding motif analysis, a custom RcisTarget database was assembled using the 'create_cistarget_motif_databases.py' tool. The construction of this database was based on the v109 Ensembl release of the Sscrofa11.1 reference genome. Feather ranking databases were constructed based on two distinct sets of regions: (1) regions encompassing 2,500 bp upstream and 500 bp downstream of each transcription start site (TSS), and (2) regions spanning 10 kb upstream and 10 kb downstream of each TSS. The binding motif list was generated by renaming human binding motifs, obtained from the SCENIC motifs' v10 public collection (https://resources.aertslab.org/cistarget/motif_collections/v10nr_clust_public/snapshots/), to their pig orthologues. High-confidence, one-to-one orthologues were obtained using the Biomart tool available on the Ensembl website.

**Cell–cell communication analysis.** Cell annotation information and raw count expression matrix was exported from Seurat, and pig gene names were converted to their human orthologues using the same process as described above. The matrix was then processed according to the standard CellChat[54] protocol using the CellChatDB.human 'secreted signalling' database.

**Comparison of datasets among mice, humans, monkeys and pigs.** To project pig single-cell data onto the mouse, human and, monkey datasets, expression matrices were exported from Seurat and gene names were converted to their human orthologues. Ensembl biomart was used to identify 14,108 high-confidence one-to-one

orthologues. All genes that were not present in all four species were excluded from the matrices. Individual matrices were then loaded into Seurat and processed individually as before using the same parameters as for pig samples. Pig, mouse and monkey datasets were then randomly down sampled so that each sample contained 25,000 cells. The transfer of cell labels between each dataset was done in a pairwise fashion using the MapQuery function in Seurat. The anchors between mouse and pig data were found with the FindTransferAnchors function (reference.reduction, 'pca'; dims, 1:50; k.filter, NA), and the function MapQuery (reference.reduction, 'pca'; reduction.model, 'umap') was used. For comparisons of cell transcriptomes, pig, monkey and mouse datasets were integrated using the IntegrateData function as before and the active identity was set to the pig cell type annotations. For identifying differentially expressed and conserved gene expression three individual objects were made from this parent object, pig-mouse, pig-monkey and mouse-monkey. The 'FindConservedMarkers' function was used to find conserved and divergent cell type-specific markers in a pairwise fashion. All comparisons used the "RNA" assay. A marker gene was classified as 'conserved' if the marker was significantly increased beyond 2-fold in the cell type of interest compared to all other cell types (adjusted p-value < 0.05) in both species tested. A gene was classified as divergent if it was significantly decreased in the cell type of interest in one species compared to all other cell types but increased in the other. To identify differentially expressed genes that include but were not limited to cell-type specific genes a cell type in each species was compared in a pairwise fashion (Epiblast 1_Pig vs Epiblast 1_Monkey, for example) using a cut-off of <0.05 for the adjusted p-value. For visualisations, the combined species matrix was scaled using the ScaleData function in Seurat.

### Generation of Figures

All figures were created in R. The following packages were predominantly used for analysis and figure creation: Seurat[65,69–71], ggplot2[72], ComplexHeatmap[73], pheatmap and monocle/monocle3[36,67,74]. The full list of packages and their versions can be found in the reporting summary.

### Statistics and Reproducibility

The statistical tests performed and biological replicates for each experiment are indicated in the figure legends.

### Reporting summary

Further information on research design is available in the Nature Portfolio Reporting Summary linked to this article.

## Data availability

The raw and processed single-cell pig gastrulation and early organogenesis data generated in this study have been deposited in the NCBI Gene Expression Omnibus (GEO) database under accession code GSE236766. The processed single-cell pig gastrulation and early organogenesis data can be viewed on our website available at https://www.nottingham.ac.uk/biosciences/people/ramiro.alberio. The mouse gastrulation and early organogenesis dataset used as a reference is available at ArrayExpress under accession code E-MTAB-6967. The monkey gastrulation dataset is available at GEO, under accession code GSE193007. The human CS7 dataset is available at ArrayExpress under accession no. E-MTAB-9388 and at GEO under accession no. GSE157329. Source data are provided with this paper.

## Code availability

No new code was created for this study however, we have deposited the scripts used for analysis and figure generation at https://github.com/DrLukeSimpson/A-single-cell-atlas-of-pig-gastrulation-as-a-resource-for-comparative-embryology.

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

## Acknowledgements

This project was supported by the Biotechnology and Biological Sciences Research Council grants [grant number BB/S000178/1] [grant number BB/T013575/1] to R.A, M.L, and J.N. We thank the staff at the School of Life Sciences Imaging facility as well as at the nMRC core imaging at the University of Nottingham, especially Jacqueline Hicks.

## Author contributions

L.S., A.S., B.P. and S.K. designed and performed experiments including scRNA-Seq, embryo dissections and wrote the paper; L.S., A.S., S.H., F.S., N.H., D.G. and T.L. performed bioinformatic analysis; A.S., T.A. and D.K. performed IF and functional experiments; M.L. supervised bioinformatic analysis; J.N. supervised the functional experiments and wrote the paper. R.A. supervised the project, designed experiments, performed dissections, and wrote the paper. All authors discussed the results and contributed to the manuscript.

## Competing interests

The authors declare no competing interests.
