## [Peer review file · Nature Communications]

A single cell atlas of pig gastrulation as a resource for comparative embryologyREVIEWER COMMENTS

Reviewer #1 (Remarks to the Author):

This research offers a single-cell transcriptomic atlas of gastrulation in pigs. It showcases the initial gastrulation process, draws comparisons across species regarding cell identities, and introduces several key perspectives:

- 1.The transition from epiblast to definite endoderm (DE) fate does not rely on EMT.
- 2.TBXT-FOXA2 double positive notochord cells have distinct origins from DE and appear later in development.
- 3.NODAL and Wnt, secreted by the primitive streak (PS) and extraembryonic endoderm, coordinate axis formation and DE differentiation.

While I concur with the authors' positions on certain issues, such as the non-existence of mesoendoderm and the role of Wnt-Nodal signaling in axis formation, I am skeptical about the manuscript's suitability for publication in Nature Communications. The novelty of the study is limited. Although pig gastrulation has been understudied, the highlighted insights on DE formation and Wnt-Nodal signaling are not groundbreaking within the context of mammalian gastrulation. Given the known conservation of these mechanisms in mice and humans, their presence in pigs is expected. Emphasizing cross-species distinctions might have added value, but the authors offer only a cursory exploration. Furthermore, the analysis and cell classification seem superficial, undermining the solidity of several conclusions. The primary figures are cluttered, presenting data that is either redundant or unclear. The manuscript also contains logical inconsistencies that need addressing. I suggest a comprehensive revision of the analysis, visuals, and narrative.

Major concerns:

- 1.Figures 2-3 align data from pigs, mice, monkeys, and humans. This alignment is problematic because of inconsistencies in cell classification across the different species' datasets. The analysis method should be uniform when integrating diverse datasets. I suggest considering the GLUE software.
- 2.Before aligning data across species, the authors should first establish corresponding time points and then align data between each.
- 3.In line 137, "pigs and mice correlated more closely to monkeys than to each other" is not solid, because the similarity may only reflect the similarities of sample stages, cell type proportions or intensities of sequencing batch effects, but not evolutionary similarity.

4. In Extended Data Fig 5b, the observed differences in pathway-related genes seem to stem from batch effects. Separate scaling for gene expression in each species is crucial.

5. I am skeptical about the borders among DE, gut, VE, AVE, yolk sac endoderm and hypoblast. The authors should not only show the heatmap of marker genes such as SOX17, FOXA1/2/3, CUBN, AFP etc. among the cell types, but also the more intuitive dotplot on the UMAP in Figure 1c, 4a, and Extend Figure 10a.

6. In figure 4d, the authors used heatmap to show classical EMT genes were absent in APS and DE. However, the DE-committed APS cells may experience a very rapid transition through EMT-MET to DE state. The intermediate cells may be very scarce that when mixed with other APS and DE cells, heatmap cannot exhibit high EMT-related genes expression. Thus, as the previous concern mentioned, the authors should show gene expression on dot plot at each time point instead of heatmap.

7. In Figure 4A and Extend Figure 10a, I found the border among subpopulations were different from the border among major populations in higher hierarchy. For example, in figure 4a, some left UMAP showed epiblast cells were reclassified into PS in the right UMAP. The authors should explain this phenomenon.

8. The SOX2+ cells visible in Figure 5c are missing in Figure 5a.

9. Figure 5e, the authors did not mark which layer was epiblast or hypoblast. Anyway, I guess the upward layer was epiblast, because it is well-known that FOXA2 is widely expressed in hypoblast. But I am very sure that I did not recognize any delamination of FOXA2+ cells in epiblast. On the contrast, the TBXT+ cells surely delaminated at the caudal side. It is totally opposite to the authors' conclusion that notochord cells delaminated later. Actually, using only FOXA2 cannot discriminate DE from VE. The authors should stain another epiblast specific marker to validate the nascent DE remains epiblast property.

10. In Figure 6a, the monocle analysis is apparently incorrect. Because APS, instead of PS1 should give rise to DE and notochord.

11. In the "Organiser-like signalling patterns of porcine cell types" part, the authors did not exactly point out which cell type was organizer.

Minor concerns:

1. Some figures, such as Figure 3, contain extraneous information.

2. In line 114, the authors said "Label projection of our pig dataset onto mouse..", but apparently, they projected mouse label onto pig in Figure 2A.

3. In line 134, "we used data projection/label transfer to apply consistent annotation to equivalent cell types for further cross-species comparisons." I cannot find the result of this analysis. Which panel showed the consistent annotations? Are the consistent annotations made based on pig or mouse or monkey?

4. In line 195, “suggesting there is a need to better define the transcriptional profiles of these for further comparisons” I can yet find the “better define of profiles”. Did the authors redefine the human and monkey ectodermal tissues?

5. In figure 2c there are two labels 24 but no label 27.

6. In figure 2-3, there are a label called NA. I understand NA means no better corresponding target, but NA is weird.

7. In line 204, there should have a “but” before “One area of controversy is that of...”

8. Figure 6d, the authors showed the mesoendoderm-like and endoderm-like labels in figure legend, but where are the cells?

9. Were the definite endoderm cluster in fig 6b fig4b and fig 1c the same? They are deadly confusing.

10. I cannot understand the line 383 sentence “Notably, the scarcity of node cells identified prior to E12.5, coupled with the fact that NODAL, CER1, LEFTY2 and DKK1 were produced by DE but not node-fated APS cells (Fig. 6e, Supplementary Table 6), suggests that the mammalian node/notochord is principally involved in secondary gastrulation.” any more. Why node did not express NODAL can postulate node is principally involved in secondary gastrulation?

11. Abbreviations such as FNS can be annotated after Epiblast -DE to improve readability in Figure 6c. The authors should improve conciseness in language.

Reviewer #2 (Remarks to the Author):

Simpson et al present an extensive resource in the manuscript entitled “A single-cell atlas of pig gastrulation as a resource for comparative embryology”

This is clearly a valuable resource to to the field of mammalian embryology to compare and contrast pig embryos with other mammals, exemplified by the comparative analysis also presented in the manuscript.

My major concern with the transcriptional analysis of mesoderm-endoderm and node trajectories is the fact that the timepoint at which this is analysed (11.5), all these cell-types are already present. This could result in misinterpretation of how the cells emerged at the earlier timepoints. This has implications to conclusions concerning shared progenitors as well as if the endoderm intercalates directly from the embryonic disc to the endoderm or through the primitive streak. For the shared progenitor it would also be important to perform the trajectory analysis, also including the mesoderm in the analysis in Figure 6.

Furthermore, I don't agree with the conclusion that the analysis in Figure 6 show that endoderm is formed from a TBXT negative epiblast progenitor. It shows that the definitive endoderm doesn't express TBXT, but it does not necessarily mean that TBXT was not expressed at an earlier stage. The trajectory in Figure 6b,d could equally well be interpreted as that FT+ cells transition within the PS1 to then split into either a APS (maintaining TBXT expression) or a DE (losing TBXT expression) trajectory. Such interpretation would also fit with the IF in Fig 5e where a small subset of FT cells is present in the streak while cells further away in the endoderm compartment are F+/T-. It is possible that some endoderm cells transition directly into the endoderm compartment through epithelial-to-epithelial transition without passing through the streak, but the data does not conclusively eliminate the possibility of a transition through a streak.

To resolve these questions, it appears that earlier samples that capture the emergence of streak, endoderm and mesoderm would be key.

The manuscript lacks information on how the embryonic disc was isolated and how extraembryonic tissues were eliminated.

It is stated that the ExE Mesoderm is identified in the earliest timepoint in alignment with formation before primitive streak formation. However, since the PS is also detected in the earliest stage, the current data neither supports nor rejects a pre-streak origin.

It is stated that amnion cells are not detected prior to E12.5 but how was amnion distinguished from surface ectoderm and neural ectoderm which was detected in the earliest stages? When is a morphologically distinct amnion emerging?

How can hypoblast be distinguished from the early definitive endoderm? From Extended Figure 1 it appears that also hypoblast is expressing FOXA2, although at a lower level. It is stated that FOXA2+/TBXT- cells emerge at E11.5, but in figure 5B they are present already at E10.5. Also, in the in silico representation and quantification of the IF, there does not appear to be any FOXA2+ cells displayed, please explain this difference.

Reviewer #3 (Remarks to the Author):

This is an important manuscript, with two parts that are well blend together. First there is a careful comparative analysis of gene expression during gastrulation in four major mammalian species. This will prove an extremely valuable resource. Then, as a derivative of this, there is a study on the origin of the endoderm that provides incontrovertible evidence that, in pig embryos, there is no common progenitor for endoderm and mesoderm and that cells fated for this germ layers arise independently from the epiblast. It has an additional point highlighting the origin of the node. There have been reports of a similar situation in mouse embryos but, as pointed out by the authors, because of the topology and developmental timing of gastrulation in the pig embryo, this study manages to provide very sound evidence for what, in places has been debated in the mouse. After this manuscript, this important feature of gastrulation can be considered a feature of mammalian development. The statement is supported by detailed immunofluorescence analysis in gastrulating pig embryos and also by functional experiments in pig and human embryonic stem cells which, in addition, provide information about the signals mediating the fate choice.

The manuscript also makes a good decision of distinguishing between 'primary' and 'secondary' gastrulation which helps clarify the fate assignments discussed.

The manuscript will make an important contribution to our understanding of early stages of mammalian development however, in places it is not clear and it would be helpful if the authors could address some issues. In particular, it is important that they do not extrapolate function from single cell analysis without empirical evidence.

There is one section I have problems with, namely the one called "organizer-like signalling patterns of porcine cell types'. The heading is correct but then there are several issues concerning references that should be corrected and also the authors should refrain from making functional statements when what they are dealing with is single cell RNA seq data.

First, ref 45 might not be the best one for the point that in mouse it has been suggested that the role of the 'classical organizer' is distributed between the node, the PS and the AVE (not the hypoblast). This is the work of P. Tam and a direct reference might be more appropriate: PMID 11566865

Also, in terms of references, neither 47 nor 48 refer to the phenotype of WNT3 mutants in human; the reference is PMID: 14872406. Also, it is not clear to me that this phenotype is similar to that of mouse Wnt3a mutants as these have a shortened trunk which is not the case in the WNT3 mutants in human.

In the discussion there is a statement: "our findings extend the understanding of the emergence of the node in mammals, which contrary to chick models⁴⁵ and earlier DE progenitors, we found no evidence for a role in A-P patterning". It is not at all clear to me how they can infer function in AP axis formation from the distribution of receptors in a UMAP. The authors should be careful with these extrapolations. They should correct this.

The expression of WNT3A in ED clusters is intriguing and important. Can they see the same thing in the human data? is this the anterior? In the mouse it is expressed during primary gastrulation in the anterior primitive streak and, later, in the Caudal Lateral Epiblast. It would be helpful if they could clarify this.

When discussing the origin of the node cells, they make the statement (line 300 onwards) “Given the APS origin of the node, and that all the cells in the APS/node trajectory do not express markers of classical EMT (Fig. 4d) this also suggests that the FTS+ cells are not mesendoderm”. This is intriguing and it appears as if these cells are closely associated with the endoderm that challenges the general belief that they are ‘mesodermal’. Could the authors comment on this? There are reports that the node has endodermal cells, but it appears from this data that it is in the same transcriptional trajectory as the endoderm.

The experiments on the origin of the endoderm with pig and human embryonic stem cells are convincing but it would be helpful if the authors could also use Wnt instead of Chiron as there are reports that the response of the cells is different for both agonists of Wnt signalling.

It would also be helpful if they could discuss how their observations relate to the data on the emergence of endoderm in hESCs and the reported role that Nodal and Wnt play on it and which differs from what their results propose. They mention this in the text but never address it

Finally, an important event in ‘secondary’ gastrulation is the appearance of the Caudal Lateral Epiblast. This is a population characterized by the appearance of the node (as they point out) and also of a population of cells coexpressing TBXT, TBX6 and SOX2. It would be helpful if the authors could pinpoint this in their data and highlight this moment in the comparative map.

**Rebuttal to reviewers**

We are grateful to the reviewers for their constructive critiques of our manuscript. We have
addressed all the reviewer's comments and added new data to our manuscript. A detailed
response to each point is provided (**in bold**) below.

**Reviewer #1 (Remarks to the Author):**

*This research offers a single-cell transcriptomic atlas of gastrulation in pigs. It showcases the*
*initial gastrulation process, draws comparisons across species regarding cell identities, and*
*introduces several key perspectives:*

*1. The transition from epiblast to definite endoderm (DE) fate does not rely on EMT.*

*2. TBXT-FOXA2 double positive notochord cells have distinct origins from DE and appear*
*later in development.*

*3. NODAL and Wnt, secreted by the primitive streak (PS) and extraembryonic endoderm,*
*coordinate axis formation and DE differentiation.*

*While I concur with the authors' positions on certain issues, such as the non-existence of*
*mesoendoderm and the role of Wnt-Nodal signaling in axis formation, I am skeptical about*
*the manuscript's suitability for publication in Nature Communications. The novelty of the*
*study is limited. Although pig gastrulation has been understudied, the highlighted insights on*
*DE formation and Wnt-Nodal signaling are not groundbreaking within the context of*
*mammalian gastrulation. Given the known conservation of these mechanisms in mice and*
*humans, their presence in pigs is expected. Emphasizing cross-species distinctions might*
*have added value, but the authors offer only a cursory exploration. Furthermore, the analysis*
*and cell classification seem superficial, undermining the solidity of several conclusions. The*
*primary figures are cluttered, presenting data that is either redundant or unclear. The*
*manuscript also contains logical inconsistencies that need addressing. I suggest a*
*comprehensive revision of the analysis, visuals, and narrative.*

**Response: We thank the reviewer for the comments on the manuscript. We agree that**
**pig gastrulation and indeed many aspects of pig development are understudied, and**
**this work offers a detailed investigation of this critical developmental time. We**
**present new insight into the emergence of DE in a flat disc embryo for the first time**
**showing direct specification of DE from epiblast cells, in the absence of**
**“mesendoderm” transition. The observations were functionally confirmed using pig**
**epiblast stem cells as well as with hESCs. The results suggest that this mechanism of**
**endoderm specification also applies to humans. These combined experiments**
**demonstrate the utility of the pig embryo for elucidating the fundamental principles of**
**flat disc embryos. We believe that this knowledge will be valuable to researchers**
**developing methods for the differentiation of human ESC into specific lineages. The**
**advantage of visualization of specific cells within the embryo offers unique insights**
**into cell-cell interactions and the topological relationships with the morphogens to**
**which they may be exposed during specification. This is not possible to model *in***
***vitro*, thus we believe complementary *in vivo* and *in vitro* investigations help gain an**
**unambiguous understanding of developmental processes. We have addressed all**
**specific concerns as requested by the reviewer.**

*Major concerns:*

*1. Figures 2-3 align data from pigs, mice, monkeys, and humans. This alignment is*
*problematic because of inconsistencies in cell classification across the different species'*
*datasets. The analysis method should be uniform when integrating diverse datasets. I*
*suggest considering the GLUE software.*

**Response: We thank the reviewer for the comment. In figures 2 and 3 we show the**
**results of reference-query mapping. In this case, because many pig cell types have**
**not been previously identified, and precisely because of inconsistencies in cell**
**classification across different species, we aimed to explore and visualise what our**
**dataset would look like using published cell annotations from other species (mouse,**
**monkey and human). We changed the wording in lines 111-113 to clarify this point.**
**We have also altered figures (figure 2a for example) from “mouse annotation” to**
**“mouse cell type annotations”. We also cite a review by (See lines 135-140; See**
**Domcke and Shendure., 2023; Cell; vol 186; p1103-1114) that explores a common**
**problem in biology, where we do not have robust definitions for many cell types.**
**Further, during development cells exist in a continuum which also presents a problem**
**for definitive labelling. Our aim was to visually show that regardless of the reference,**
**there is large agreement in the annotations of many cell types. For example, what a**
**human and a mouse embryologist call an epiblast cell we also call an epiblast cell and**
**so on. We also highlight cell types where the annotations differ depending on the**
**reference (See lines 120-122 and 124-128). What the authors of Pijuan-Sala et al (2019)**
**refer to as “mesenchyme” for example we, Zhai et al (2023; Nature, 612, 732-738) and**
**Tyser et al (2021, Nature, 600, 285-289) would call extraembryonic mesoderm/yolk-sac**
**mesoderm. We posit that our cell annotations are a reference point and we provide a**
**table of all markers used for cell type identification and their source publications (See**
**Supplementary Data 1, formerly Supplementary Table 1), something that is often**
**missing from many publications utilising single-cell datasets. For any comparative**
**analyses between cell types across species we used reference-query mapping to**
**apply a single uniform annotation (our annotations) to datasets prior to comparisons.**
**We have reworded lines 130-134 and this is also stated explicitly in the methods**
**section (See lines 741 to 769).**

**Regarding the use of GLUE software, we disagree that it offers any advantage to what**
**we used. GLUE is another tool for dataset integration, and we have already performed**
**dataset integration prior to any comparative analyses. After applying a uniform**
**annotation, samples were pre-processed individually and then integrated using**
**Seurat’s native integration function (IntegrateData). As a brief outline of the**
**integration steps, we identified highly variable features within the species datasets.**
**High-variance genes (or features) are typically selected because they are the ones**
**that most likely define cell types and states. We used Reciprocal Principal Component**
**Analysis (RPCA) during the anchor finding, to reduce the complexity of the data and**
**preserve the most important variation between cells. This identifies shared principal**
**components across datasets and thus finds the fundamental similarities between**
**different datasets, despite any batch effects or other noise. After dimensionality**
**reduction, Seurat constructs a shared-nearest-neighbour (SNN) graph, which is used**
**to identify anchors. Because the datasets have been “harmonized” through RPCA,**
**cells of similar types from different datasets should now be closer to each other in**
**low-dimensional space. Seurat identifies pairs or clusters of cells from different**
**datasets that are “close together” in this space as potential anchors. Seurat assigns**
**scores to these potential anchors based on the strength of their similarities and filters**
**out weaker, less certain anchors. This leaves a set of high-confidence anchors that**
**represent strong, biologically meaningful similarities between the datasets. This is**
**used to integrate data and correct expression values between datasets. Seurat is**
**perhaps the most commonly used package for analysing single cell-RNA seq data and**

for integration of single-cell datasets. In a recent study it was shown to be one of the
best tools for both batch correction (See Tran et al., 2020, *Genome Biology*, 21, 1-32)
and multi-species integration (See Song et al., 2023; *Nature Communications*, 14,
6495). This method of integration has been employed for dataset integration in
numerous studies and for cross-species comparisons (See Zhai et al., 2023; *Nature*,
612, 732-738) as a recent example). Seurat also has the added benefit of creating
separate “RNA” and “integrated” (Batch-corrected) assay slots allowing analyses to
be performed on both the uncorrected values as well as the “corrected” values based
on the results of the integration, a feature not offered by many other packages.

*2. Before aligning data across species, the authors should first establish corresponding time*
*points and then align data between each.*

**Response:** We thank the reviewer for their comment. In the case of our quantitative
comparisons between mouse, monkey and pig all three datasets represent samples
from the beginning of gastrulation through to approximately the 10-somite stage (Fig.
1a). While morphologically these samples are approximated to be equivalent, the
rationale behind these comparisons is that we do not know which stages are
“equivalent” and this may also differ on a cell type basis. Given that the reference-
query mapping finds equivalent cell states based on which is the most
transcriptionally similar, having multiple, differently staged samples is useful.
Previous studies have performed similar types of analyses (See Tyser et al., 2021,
*Nature*, 600, 285-289; Zhai et al., 2023; *Nature*, 612, 732-738; Ton et al., 2023, *Nature*
*Cell Biology*, 1-12). Importantly, these studies also show that cell-type equivalents are
not always found in morphologically equivalent stages. We also find evidence of
these previously described discrepancies, as well as some that have not previously
been reported.

*3. In line 137, “pigs and mice correlated more closely to monkeys than to each other” is not*
*solid, because the similarity may only reflect the similarities of sample stages, cell type*
*proportions or intensities of sequencing batch effects, but not evolutionary similarity.*

**Response:** We agree with the reviewer, and we only meant to describe the results of
our correlation of the entire datasets. We agree that various factors may affect the
correlation of samples. We did not infer evolutionary similarity. We removed this
statement from the paper and the corresponding figure.

Regarding batch effects (mentioned in this comment and point 4, below), we accept
the mouse, monkey, pig embryo collections have been performed by different labs
and processed for sequencing independently and thus some batch effects are
expected. However, all three studies utilised the 10X Chromium system and the
stages covered by each atlas are approximately equivalent (See our response to point
2). Given that in all three studies, each sample also represents a particular
developmental stage, batch correction, either via logistic regression or through
Seurat’s native integration function would likely remove real biological variation.
While we utilised integration for finding equivalent cell types across datasets and
visualisation it is not recommended to use this assay for comparative analyses (This
has been stated by the developers of Seurat:
<https://github.com/satijalab/seurat/issues/1717>). Given that it is currently unfeasible
for any single lab to perform a time course of pigs, mice and monkeys, this remains
the best means of comparing these stages of development to date. While one can
rarely rule out batch effects completely, we have generally reported large differences
in gene expression, and we maintain that our reported findings likely reflect real
transcriptional differences. While we hope that future work will attempt to replicate,

validate and extend our findings, we believe that these comparisons will be of great
value to the scientific community and are expected in a paper of this kind. Further, the
practice of integrating independently created datasets from different species for
means of cell-type comparison is very common. Indeed, this has been implemented in
numerous high-profile papers including (See Tyser et al., 2021, Nature, 600, 285-289;
Zhai et al., 2023; Nature, 612, 732-738; Ton et al., 2023, Nature Cell Biology, 1-12;
Mayshar et al., 2023, Cell, 186, 2610-2627) to name a few.

*4. In Extended Data Fig 5b, the observed differences in pathway-related genes seem to stem*
*from batch effects. Separate scaling for gene expression in each species is crucial.*

**Response:** Given that these species have been sequenced by different labs and using
two different sequencing technologies batch effects are possible but as discussed
above it is currently unfeasible for a single lab to sequence embryos of these kinds
alone. Regarding the scaling of the values, we want to clarify that the data presented
has first normalised, i.e. the counts of each gene in each individual cell have first
been divided by the total number of counts for that cell and multiplied by 10,000.
These values +1 are then log-transformed (See NormalizeData function in Seurat).
Thus, each cell regardless of species has been normalised in a way that mitigates the
effects of sequencing depth and other technical biases and gene expression is
comparable. We used this normalised data to find differentially expressed genes
between species and therefore data scaling would not affect the identification of these
genes. Furthermore, the total number of genes in each species dataset is similar and
we excluded any genes that had a count of zero in any one species to avoid
comparing genes that have “dropped out” in one species with lowly expressed genes
in another.

Regarding the scaling of the data being performed individually, the ScaleData
function is performed on the combined, normalised, pig-mouse-monkey matrix
whereby the mean gene value (across all cell types and species) is subtracted from
each (normalised) gene value in each cell. These scaled values are then divided by
the standard deviation of the gene. Scaling is generally only useful when a gene has
differential expression within the data being scaled. Therefore, scaling genes across
species is useful to visualise expression differences between species in both cell-
type-specific genes and non-cell-type-specific genes. Further, genes which differ
greatly in their “absolute expression levels” can be displayed together. Scaling the
genes on species individually would be problematic for identifying differences
between species in genes that are lowly but ubiquitously expressed, as is the case
with many of the genes displayed in Extended Data Fig.5a. As an example, the gene
FLNA (See rebuttal Fig. 1 below) while expressed in most cell types, it has far lower
counts in mice embryos than in pig or monkey (a). This is also true after normalising
for sequencing depth (b) and this trend is also clear when scaling the combined
species data (c). However, when scaling each species individually, this difference is
not visible (d), nor is it appropriate to compare between species in this way. To make
this clearer for readers we have rewritten sections of the methods section (See lines
749 to 769) to add detail and clarify the steps taken in the analysis and what data is
displayed. We have also clarified what data is displayed in each figure legends.

Fig. 1. Scaling gene expression across species enables the accurate representation of absolute differences in gene expression after accounting for variations in sequencing depth. This approach allows the visualisation of genes that exhibit intraspecies variability as well as genes that are consistent within a species but display variability when compared across different species. **a.** (Left) raw counts for POU5F1, a gene with conserved gene expression profiles that vary between cell types within each species. (Right) raw counts for FLNA a gene that is expressed in most cell types but varies between species. **b.** Expression of POU5F1 and FLNA after normalising for sequencing depth. Note that even after normalisation there is considerably less expression of FLNA in mice. **c.** Scaled gene expression of POU5F1 and FLNA, scaling expression on the combined, normalised pig-mouse-monkey matrix. Note that scaling across species reflects the differences seen in FLNA expression both in the raw counts and after adjusting counts for sequencing depth. **d.** Scaled expression of POU5F1 and FLNA when data scaling is performed on each species independently. Note that individual scaling results in the expression values of FLNA no longer reflect the absolute differences in transcript abundance.

*5. I am skeptical about the borders among DE, gut, VE, AVE, yolk sac endoderm and*
*hypoblast. The authors should not only show the heatmap of marker genes such as SOX17,*
*FOXA1/2/3, CUBN, AFP etc. among the cell types, but also the more intuitive dotplot on the*
*UMAP in Figure 1c, 4a, and Extend Figure 10a.*

**Response:** The borders of these clusters have been determined by performing
clustering using the FindClusters function in Seurat on the definitive endoderm,
hypoblast/gut and extra-embryonic endoderm “macro” clusters after subsetting from
the main object (visible in Fig 1c) using a resolution of 0.5. We have detailed our
reasoning for performing clustering on multiple “macro” clusters in our response to
point 7 below. While we agree that feature plots (UMAPs in which each cell is
coloured by the gene expression values) can often be more intuitive clusters often
overlap in low-dimensional 2D plots as a result it is often difficult to see where one
cluster ends and another begins. Further, these plots take up a lot of space in the
case of Figure 1c, 4a, and Extend Data Fig. 10a, this would amount to more than 100
marker plots. To facilitate ease of interpretation for the reader we have provided the
requested and some additional feature plots for these figures in Supplementary Fig 1-
3. However, given that many of the genes used to distinguish endodermal subclusters
are expressed in non-endodermal subtypes we have used the subclustered endoderm
UMAPs. For additional clarity, the heatmap in Extended Data Fig. 10d has been
replaced with a heatmap only showing diagnostic markers. Newly discovered markers
of these groups are now available in Supplementary Data 9 and 10 (formerly
Supplementary Table 8 and 9).

*6. In figure 4d, the authors used heatmap to show classical EMT genes were absent in APS*
*and DE. However, the DE-committed APS cells may experience a very rapid transition*
*through EMT-MET to DE state. The intermediate cells may be very scarce that when mixed*
*with other APS and DE cells, heatmap cannot exhibit high EMT-related genes expression.*
*Thus, as the previous concern mentioned, the authors should show gene expression on dot*
*plot at each time point instead of heatmap.*

**Response:** We thank the reviewer for their comment; our dataset includes 24,874 cells
between epiblast, PS, APS, nascent mesoderm and DE clusters. Indeed, in the UMAP
or in pseudo-time plots, there are no gaps suggesting that our sampling was
sufficient to capture transient cell states. We broadly observed no significant
upregulation of classical EMT markers even in these fleeting populations. Given that
the heatmap in Fig.4d shows expression of individual cells relative to other cells and
that the majority of cells have no detectable expression even moderate expression
can be visible while crowding can indeed be an issue. This is often exacerbated with
feature plots as dots may overlap, obscuring information. To address the reviewer's
concerns we have included a supplementary figure (Supplementary Fig 1) showing
feature plots of EMT markers.

*7. In Figure 4A and Extend Figure 10a, I found the border among subpopulations were*
*different from the border among major populations in higher hierarchy. For example, in figure*
*4a, some left UMAP showed epiblast cells were reclassified into PS in the right UMAP. The*
*authors should explain this phenomenon.*

**Response:** The reason this is the case is that Fig 4a shows the isolated clusters of
interest, their labels have been given based on average expression of marker genes
across the cluster. Inspection of these cells revealed heterogeneity. Therefore, in new
Fig 4b we show the same cells at higher clustering resolution, and thus we can

identify sub-populations. It is worth noting that the borders are dependent on the
clustering resolution and that each point in the UMAP represents a transcriptional
state along a continuum. To the reviewer's point, another way to achieve this is to
sub-cluster a single "macro" cluster of interest in isolation; for example, we could
have subclustered only the APS to identify node-like cells and kept the other clusters
unaltered. However, we did not do this as it would impart a bias on the data, resolving
heterogeneity in one cluster and ignoring possibly relevant heterogeneity in other
neighbouring clusters. If we subcluster all the original macro-clusters, this results in a
very large number of clusters that would hinder meaningful interpretation. As an
example, using our current approach we identified cells that did not appear to have a
definitive Epiblast, APS or PS identity (Formerly group PS1 in Fig. 4b). We believe that
these cells are representative of the early caudal epiblast. If we had sub-clustered just
the APS we would have not identified this population, similarly applying a high
enough resolution to find this group on our entire dataset would result in over 60
clusters. However, our two-scale approach offers increased granularity of cell
identities without resulting in an unmanageable number of clusters that would not
have otherwise been possible. Indeed, using multiple clustering resolutions is a
relatively common approach allowing us to make observations at different levels of
granularity, describing both "the forest" and "the trees".

*8. The SOX2+ cells visible in Figure 5c are missing in Figure 5a.*

**Response:** The SOX2 staining has now been added to 6a (formerly fig5a; coloured
violet).

*9. Figure 5e, the authors did not mark which layer was epiblast or hypoblast. Anyway, I guess
the upward layer was epiblast, because it is well-known that FOXA2 is widely expressed in
hypoblast. But I am very sure that I did not recognize any delamination of FOXA2+ cells in
epiblast. On the contrast, the TBXT+ cells surely delaminated at the caudal side. It is totally
opposite to the authors' conclusion that notochord cells delaminated later. Actually, using
only FOXA2 cannot discriminate DE from VE. The authors should stain another epiblast
specific marker to validate the nascent DE remains epiblast property.*

**Response:** We apologize for the oversight; we have labelled Epi and Hypo in Fig. 6d
top left (formerly Fig. 5d). We also included new images (Fig. 6e) showing confocal
sections of embryos stained with NANOG (an epiblast marker which is maintained in
DE progenitors/nascent DE), FOXA2 and TBXT antibodies. We identified and labelled
(arrowheads) early NANOG-positive definitive endoderm cells scattered among
NANOG-negative, FOXA2 positive cells in the hypoblast layer. Notably, these cells are
readily detectable in E11.5 embryos while few are detectable in E12 embryos. This is
consistent with the scRNAseq data presented in Fig. 4 and 5 (formerly Fig. 6).

*10. In Figure 6a, the monocle analysis is apparently incorrect. Because APS, instead of PS1
should give rise to DE and notochord.*

**Response:** We thank the reviewer for their comment. While this figure has since been
replaced (in line with comments from other reviewers) Monocle analysis did indeed
show that both PS1 (which we believe represent early caudal epiblast cells, as
opposed to the posterior-most epiblast cells, which are fated to mesoderm) and APS
are fated to DE. Generally, most PS1 cells are located at the start of the trajectory and
represent cells that have not committed to either fate and generally APS cells are
located along branches of the trajectory indicating that they are biased toward either
notochord or definitive endoderm, with some also appearing on the uncommitted
branch. This is the general trend, however, there are some exceptions, notably some

PS1 cells are located on the DE fated branch. The latest version of this figure (now
Fig. 5a) shows the cell trajectories overlayed onto the UMAP plot. As before this
shows that caudal epiblast cells bifurcate committing toward either APS or
mesodermal fates, APS cells themselves then bifurcate toward DE or node fates.

*11. In the “Organiser-like signalling patterns of porcine cell types” part, the authors did not*
*exactly point out which cell type was organizer.*

**Response:** While we comment on organiser-like signalling patterns of our subgroups,
we do not think that any one cell type fits the “traditional” definition of organiser as it
exists in amphibians. Indeed, a review on the mammalian organiser (See Arias and
Steventon., 2018; *Development*, 145, 5) and work carried out by P. Tam (See Kinder et
al., 2001; *Development*, 128, 3623-3624; Tam & Steiner., 1999; 126, 5171-5179) whose
work suggest that functions associated with the amphibian organiser (such as the
establishment of the A-P axis, axial extension, etc.) are split between distinct
embryonic cell types. While we cannot infer functionality directly (as reviewer 3 points
out; see comment below), the gene expression profiles in our data are consistent with
this idea. For example, WNT3 is required for the formation of an A-P axis as it
specifies the primitive streak (Liu et al., 1999; *Nature Genetics*, 22, 361-365). Our
staining data shows the first TBXT+ cells in the posterior emerge between E10.5 and
E11.5. We detect TBXT+ cells of the PS and nascent mesoderm expressing WNT3,
consistent with the data in mice. At E11.5, we find cells with an APS-like expression
profile (POU5F1+, EOMES+ FOXA2+, TBXT-) that also express genes associated with
the establishment of A-P patterning (such as the WNT-BMP-NODAL inhibitor CER1),
and the WNT signalling pathway inhibitor DKK1 and TGF-beta inhibitor LEFTY2.
Further, these same cells express high levels of NODAL, another known factor
involved in the establishment of the body axis. Based on these findings we suggest
that the transcriptional profiles of these cells (which our monocle analysis suggests
are endoderm-fated) are consistent with a role in axis establishment. These genes are
also expressed in anterior hypoblast-like cells (SOX17+, OTX2+, APOE+). By contrast,
E12.5-E15 APS-like cells (POU5F1+, FOXA2+, TBXT+) fated to node and node cells
(FOXA2+, TBXT+, CHRD+) do not express CER1, LEFTY2 or DKK1, but instead the
axial patterning gene SHH. Given that axis elongation occurs around this same period
these cells’ transcriptional profiles are consistent with a role in axial elongation. Since
node cells are not present during the period where axes are established, it suggests
node cells are not involved in axis specification. Together, our results are consistent
with the idea that the axis establishment aspect of the organiser function is split
between the anterior hypoblast, the endoderm-fated APS and the mesoderm-fated
primitive streak, and the axis elongation aspect of the organiser function is carried
out by the node/notochord. Therefore, it does not seem appropriate in this context to
refer to any one cell type as “organiser”. We have discussed this in the revised ms.
L428-446.

*Minor concerns:*

*1. Some figures, such as Figure 3, contain extraneous information.*

**Response:** We agree with the reviewer that figure 3 is extensive and contains groups
of cells with very few overlaps between the queried cell types from humans. We have
removed many of the groups to increase the clarity of this figure.

*2. In line 114, the authors said “Label projection of our pig dataset onto mouse..”, but*
*apparently, they projected mouse label onto pig in Figure 2A.*

**Response: We initially mapped our pig data to mouse, monkey and human reference**
**datasets and then transferred their respective annotations onto our pig dataset to**
**compare cell annotations (discussed further in comment 1 above). After establishing**
**that our annotations offer some advantages in terms of cell classification over the**
**mouse and monkey datasets (See lines 111 to 128), we then used our pig annotations**
**for any subsequent analysis (See lines 130 to 134). As this was unclear, we have**
**altered these sentences for clarity and included UMAPs showing the results of our pig**
**annotations on the monkey and mouse datasets in Fig. 2e.**

*3. In line 134, “we used data projection/label transfer to apply consistent annotation to*
*equivalent cell types for further cross-species comparisons.” I cannot find the result of this*
*analysis. Which panel showed the consistent annotations? Are the consistent annotations*
*made based on pig or mouse or monkey?*

**Response: Please see the response above. We have now included UMAPs showing**
**the results of using our pig annotations on the monkey and mouse datasets following**
**alignments in Fig. 2e.**

*4. In line 195, “suggesting there is a need to better define the transcriptional profiles of these*
*for further comparisons” I can yet find the “better define of profiles”. Did the authors redefine*
*the human and monkey ectodermal tissues?*

**Response: While we have provided a referenced list of all the diagnostic markers**
**used in our study (See Supplementary Data 1, formerly Supplementary Table 1), a**
**recent publication discusses the need to standardise and better define cell types from**
**these type of datasets (See Domcke & Shendure., 2023; Cell, 186, 1103-1114). We used**
**diagnostic markers and provided a list of all the significant cell type marker genes in**
**our clusters. Regarding whether we redefined human and monkey ectodermal tissues,**
**in lines 120-121 we discuss that our amnion and TE cluster would simply come under**
**the bracket of “surface ectoderm” using mouse annotations (Pijuan-Sala et al., 2018).**
**Using monkey annotations (Zhai et al., 2022) would result in the amnion also being**
**labelled as “surface ectoderm 1” and a different surface ectoderm cluster would be**
**labelled amnion. However, using human annotations (Tyser et al., 2021) would result**
**in the labelling of the cluster as amnion (Extended Data Fig. 2c&d) which we feel is**
**more appropriate given the high expression of amnion markers ISL1, TFAP2C and**
**GATA3 (Extended Data Fig. 1e). Given that Pijuan-Sala only provides a heatmap of**
**diagnostic markers it appears that no amnion markers were investigated, despite**
**evidence that by the end of their time course amniogenesis would have begun (See**
**Pereira et al., 2011; BMC developmental biology, 11, 1-13). Further, it was not reported**
**which markers were used in Zhai et al. to make the distinction between surface**
**ectoderm and amnion. Indeed, marker genes GABRP and HEY1 (See Extended Data**
**Fig.1e; Supplementary Data 1&2 formerly Supplementary Table 1&2) are also**
**consistent with the late primate amnion profile (See Tyser et al., 2021; Rostovskaya et**
**al., 2022; Cell Stem Cell, 29, 5, 744-759). Given that using these markers resulted in an**
**agreement between our annotations and that of Tyser et al. (who also provided a list**
**of amnion markers), we conclude that our annotation is correct. We then applied our**
**annotations to the mouse and monkey datasets for all cell type comparisons (the**
**results of this alignment can be seen in Fig 2e. The results of the comparisons**
**between pig and the “redefined” monkey amnion, anterior surface ectoderm and**
**posterior surface ectoderm can be found in Supplementary Data 5-7 (formerly**
**Supplementary Tables 3-5) along with all other aligned cell types. We have also added**
**GABRP to the marker heatmap in Extended Data Fig.1e to make it easier for the reader**
**to see this information.**

*5. In figure 2c there are two labels 24 but no label 27.*

**Response: We thank the reviewer for bringing this to our attention, this has been**
**corrected.**

*6. In figure 2-3, there are a label called NA. I understand NA means no better corresponding*
*target, but NA is weird.*

**Response: In this case, NA is the label given to a number of cells by Pijuan-Sala and**
**colleagues. While the authors have not detailed these cells in their publication**
**presumably these cells do not fit any known cell type and were not excluded through**
**standard QC filtering. These cells were also presumably excluded from the UMAP plot**
**as they have no UMAP coordinates. To address the reviewers concern we have**
**changed this to “unknown cell type”.**

*7. In line 204, there should have a “but” before “One area of controversy is that of...”*

**Response: We have corrected this in line 210.**

*8. Figure 6d, the authors showed the mesoendoderm-like and endoderm-like labels in figure*
*legend, but where are the cells?*

**Response: This label refers to mesendoderm-like cells only in the sense that these**
**cells have detectable transcripts of FOXA2, TBXT and (low) SOX17, however, the vast**
**majority of these cells are part of the node cluster (Fig 6g, right) as such their**
**transcriptional profiles are nearly identical to SOX17- cells within the same group**
**(Extended Data 6). As detailed in the manuscript we do not think they fit the definition**
**of mesendoderm, thus, we have removed this label to avoid confusion replacing it**
**with the term SOX17+ node.**

*9. Were the definite endoderm cluster in fig 6b fig4b and fig 1c the same? They are deadly*
*confusing.*

**Response: Yes, in all three figures the definitive endoderm cluster is the same group**
**of cells. While in Fig 1c their UMAP coordinates are calculated based on the first 25**
**principal components of our entire dataset, Fig 4b’s UMAP coordinates are based on**
**the first 25 principal components of the epiblast, APS, PS and nascent mesoderm**
**clusters only. In Fig 6b their UMAP coordinates are calculated from the first 25**
**principal components of the PS1, APS, Node and DE subclusters only. In each case,**
**the exclusion or inclusion of other cell types will impact the calculated coordinates of**
**the cells in the UMAP. This facilitates a two-dimensional visualisation of the variance**
**of particular populations and can show local structure in the data as opposed to more**
**global structures. This approach is standard in many papers using dimensionality**
**reduction techniques. For example, Zhai et al., 2023; Nature, 612, 732-738.**

*10. I cannot understand the line 383 sentence “Notably, the scarcity of node cells identified*
*prior to E12.5, coupled with the fact that NODAL, CER1, LEFTY2 and DKK1 were produced*
*by DE but not node-fated APS cells (Fig. 6e, SupplementaryTable 6), suggests that the*
*mammalian node/notochord is principally involved in secondary gastrulation.” any more.*
*Why node did not express NODAL can postulate node is principally involved in secondary*
*gastrulation?*

**Response:** Regarding the rationale for this statement, we find a much greater
abundance of DE-fated APS cells at the very start of gastrulation that express NODAL,
CER1, LEFTY2 and DKK1. The encoded proteins from these genes have functions in
the establishment of A-P patterning (the defining feature of primary gastrulation; See
Arias and Steventon., 2018; Development, 145, 5) by creating morphogen gradients.
Given that there are few node cells at the start of gastrulation (3 cells between E11.5
and E12, 4 samples in total) and that we do not find expression of these morphogens
in the node-fated cells of the APS (of which there are 19 cells between E11.5-E12) this
suggests that the node itself cannot play an inductive role in this process. Further,
given the rapid accumulation of node-fated APS cells and Node cells after E12 and the
expression of factors relating to axial extension (the defining characteristic of
secondary gastrulation; Arias and Steventon., 2018) in these groups, we conclude
that the node is principally involved in secondary gastrulation. As to why the node-
fated and node cells did not express NODAL, and the DE-fated cells did, we cannot
say with certainty however no NODAL was detected in monkey or human node/axial
mesoderm, respectively.

*11.Abbreviations such as FNS can be annotated after Epiblast -DE to improve readability in*
*Figure6c. The authors should improve conciseness in language.*

**Response:** We have changed the original figure. For consistency, we left the
abbreviations as described in the text, so that the reader can follow the pattern of cell
distribution based on the expression of the different markers. We feel Epiblast-DE in
the figure would be at odds with the other cells which are defined by the genes they
express. Further, there are exceptions, for example, while all FNS+ cells are endoderm
fated, NANOG was not detected in every DE-fated cell.

*Reviewer #2 (Remarks to the Author):*

*Simpson et al present an extensive resource in the manuscript entitled “A single-cell atlas of*
*pig gastrulation as a resource for comparative embryology”*

*This is clearly a valuable resource to the field of mammalian embryology to compare and*
*contrast pig embryos with other mammals, exemplified by the comparative analysis also*
*presented in the manuscript.*

*My major concern with the transcriptional analysis of mesoderm-endoderm and node*
*trajectories is the fact that the timepoint at which this is analysed (11.5), all these cell-types*
*are already present. This could result in misinterpretation of how the cells emerged at the*
*earlier timepoints. This has implications to conclusions concerning shared progenitors as*
*well as if the endoderm intercalates directly from the embryonic disc to the endoderm or*
*through the primitive streak. For the shared progenitor it would also be important to perform*
*the trajectory analysis, also including the mesoderm in the analysis in Figure 6.*

**Regarding the use of earlier time points, we have previously published single-cell**
**data of E5 to E11 of pig development (See Ramos-Ibeas et al., 2019; Nature**
**Communications; 10; 500) which was recently complemented with larger-scale**
**dataset of E9 and E11 pig embryos (See Dufour et al., 2023; BioRxiv) which show no**
**expression of mesoderm genes (e.g.TBXT). We re-analysed the Dufour et al., dataset**
**(rebuttal figure 3) and identified a cluster originally labelled as “mesendoderm” to be**
**mislabelled in the preprint, as these cells have close identity to AVE cells (e.g. CER1,**
**LEFTY and GSC). While the reviewer is correct that even at our two earliest stages**
**(E11.5) we do find endodermal cells, they make up a small percentage of the total**
**cells at this time point (2.5%, 236 cells) these cells therefore represent the earliest**

cells committing to this fate. As mentioned above we only observe 3 node cells
between E11.5-12 and 19 node-progenitors (Determined using pseudotime analysis).
As to whether we have missed transient intermediate progenitors, in our UMAP and
pseudo-time plots, we observe no “gaps” suggesting that our sampling was sufficient
to capture all transient cell states between epiblast and APS, and between APS and
DE or node. Further, the samples taken at E11.5-E12.5 represent pools of embryos
and while they are morphologically very similar, they likely represent a range of time-
points around E11.5. Given the lack of evidence of meso/endoderm cells at earlier
time points and that practically it is not possible to sample embryos at less than half-
482 day intervals, we believe that our sampling intervals have captured the relevant cell
types present in pig embryos at the start of gastrulation. Regarding the reviewer’s
suggestion to include mesodermal fated cells in the trajectory analysis, we have now
redone trajectory analysis using monocle3 with these cells and we showing factors
that may influence mesoderm fate included (See Fig.5 and Extended Data Fig.

Fig. 2. Feature plots of cell type markers in E5, E7, E9 and E11 pig embryos (data from Dufour et al., 2023). These data suggest that the labels used by Dufour et al., 2023 are correct with the exception of the cluster initially labelled as “Mesoderm” which groups with hypoblast cluster, expresses high levels of hypoblast markers HNF1B, NID2, GATA4 and COL4A2. Given the additional expression of anterior hypoblast markers this group is very likely to represent anterior hypoblast/AVE.

Fig. 3. Co-expression of key mesodermal and endodermal genes in E5, E7, E9 and E11 pig embryos shown in Fig. 2 (data from Dufour et al., 2023). We find no evidence of mesendoderm-like cells in any early pig cells at E5-E11 as evidenced by co-expression of TBXT and FOXA2, TBXT and SOX17 or TBXT and SNAI1.

*Furthermore, I don't agree with the conclusion that the analysis in Figure 6 show that*
*endoderm is formed from a TBXT negative epiblast progenitor. It shows that the definitive*
*endoderm doesn't express TBXT, but it does not necessarily mean that TBXT was not*
*expressed at an earlier stage. The trajectory in Figure 6b,d could equally well be interpreted*
*as that FT+ cells transition within the PS1 to then split into either a APS (maintaining TBXT*
*expression) or a DE (losing TBXT expression) trajectory. Such interpretation would also fit*
*with the IF in Fig 5e where a small subset of FT cells is present in the streak while cells*
*further away in the endoderm compartment are F+/T-. It is possible that some endoderm*
*cells transition directly into the endoderm compartment through epithelial-to-epithelial*
*transition without passing through the streak, but the data does not conclusively eliminate*
*the possibility of a transition through a streak.*

*To resolve these questions, it appears that earlier samples that capture the emergence of*
*streak, endoderm and mesoderm would be key.*

**Response:** Please see our answer to the previous comment regarding sampling of
**earlier time points.** As to whether the endoderm progenitor is TBXT-negative, it is true
**that a small number of DE fated cells have detectable TBXT expression, the levels of**
**TBXT expression are less than half that of mesoderm fated cells and less than a**
**quarter of the levels associated with node fated cells (Fig. 4d, Fig. 5c; rebuttal table 1**
**and table 2).** However, the fact that the majority of endoderm-fated APS cells have no

detectable TBXT expression and that we observe no TBXT protein expression in these
 same cells suggests that definitive endoderm primarily forms from TBXT-negative
 cells. Furthermore, we have never detected TBXT+ cells in the endoderm layer after
 immunostaining.

Cell Type Trajectory	TBXT NULL (%)	TBXT DETECTABLE (%)	FOXA2 NULL (%)	FOXA2 DETECTABLE (%)	TBXT AND FOXA2 DETECTABLE (%)	TBXT ONLY DETECTED (%)	FOXA2 ONLY DETECTED (%)
Primitive Streak 1 - Endoderm fated	100	0	95.23809524	4.761904762	0	0	4.761904762
Primitive Streak 1 - Node fated	100	0	50	50	0	0	50
Primitive Streak 1 - Uncommitted	95.85308057	4.146919431	86.07819905	13.92180095	0.888625592	3.258293839	13.03317536
Anterior Primitive Streak - Endoderm fated	86.17886179	13.82113821	25.20325203	74.79674797	12.19512195	1.62601626	62.60162602
Anterior Primitive Streak - Node fated	48.93964111	51.06035889	9.298531811	90.70146819	47.63458401	3.425774878	43.06688418
Anterior Primitive Streak - Uncommitted	71.65048544	28.34951456	23.88349515	76.11650485	22.52427184	5.825242718	53.59223301
Definitive Endoderm	98.58242203	1.417577967	37.74807614	62.25192386	1.134062373	0.283515593	61.11786148
Node	43.88646288	56.11353712	5.676855895	94.3231441	53.27510917	2.838427948	41.04803493

**Table 1.** Percentages of cells with detectable TBXT and FOXA2 (Based on 5896 cells
 between PS1, APS, Node and Definitive Endoderm clusters). The majority of endoderm-
 fated cells and their uncommitted progenitors have no detectable TBXT expression.

Cell Type Trajectory	Median TBXT expression (all cells in group)	Median FOXA2 expression (all cells in group)	Median TBXT (positive cells only)	Median FOXA2 expression (positive cells only)
Primitive Streak 1_Endoderm fated	0	0	NA	0.580139249
Primitive Streak 1_Node fated	0	0.599848256	NA	1.431802389
Primitive Streak 1_Uncommitted	0	0	0.485266595	0.742566473
Anterior Primitive Streak_Endoderm fated	0	1.024366974	0.408267946	1.246220148
Anterior Primitive Streak_Node fated	0.227341385	1.576863946	0.852821465	1.656229572
Anterior Primitive Streak_Uncommitted	0	0.889430461	0.438241212	1.129043835
Definitive Endoderm	0	0.674862291	0.566066199	0.934364436
Node	0.603535231	1.923536215	1.09661628	1.971254021

**Table 2.** Median gene expression values based on 5896 cells between PS1, APS, Node and
 Definitive Endoderm clusters. We observe that cells fated for the endoderm and those
 uncommitted exhibited a median TBXT expression of zero. Notably, the small subset of
 TBXT-expressing cells within the uncommitted and endoderm-fated populations
 demonstrated a median expression level of TBXT that was approximately half that observed
 in cells fated for the node.

*The manuscript lacks information on how the embryonic disc was isolated and how*
*extraembryonic tissues were eliminated.*

**Response: We thank the reviewer for bringing this to our attention. The embryos were**
**dissected manually using needles and extraembryonic tissues were cut near the**
**embryonic disc. Part of the trophectoderm and hypoblast layer was maintained in the**
**samples deliberately to avoid losing any cells from the atlas. We added a sentence to**
**describe this in the methods section (lines 596-600).**

*It is stated that the ExE Mesoderm is identified in the earliest timepoint in alignment with*
*formation before primitive streak formation. However, since the PS is also detected in the*
*earliest stage, the current data neither supports nor rejects a pre-streak origin.*

**Response: We agree with the reviewer's comment and have removed mention of this**
**from the manuscript.**

*It is stated that amnion cells are not detected prior to E12.5 but how was amnion*
*distinguished from surface ectoderm and neural ectoderm which was detected in the earliest*
*stages? When is a morphologically distinct amnion emerging?*

**Response: The neural ectoderm cluster was identified based on having higher**
**expression of genes such as PTN and CRABP1 and lower expression of pluripotency**
**genes NANOG and POU5F1. However, while this is true across the whole data set this**
**trend is the opposite at the earliest time points (E11.5 and E12). While we maintain**
**that later this cluster acquires a neural character this label may not have been**
**appropriate overall. Therefore, we have changed the name of this cluster to "Epiblast**
**4". Regarding the surface ectoderm and amnion clusters, the annotation of amnion**
**was primarily based on high expression of ISL1 and TFAP2A as well as**
**extraembryonic marker GATA3. Surface ectoderm cluster annotation was based on**
**higher expression of TFAP2C, and GRHL1&2 and reduced expression of TFAP2A,**
**ISL1 and GATA3. Consistent with an embryonic location the posterior surface**
**ectoderm cluster expresses posterior HOX genes (See Supplementary Data 2,**
**formerly Supplementary data 2). Furthermore, the amnion cluster is annotated as**
**such when mapping to human annotations and has no detectable HOX gene**
**expression, consistent with an extra-embryonic identity. While we detect amnion cells**
**from E12.5 onward the amnion is not morphologically distinct until around E13 (See**
**Patten., 1927; The Embryology of the Pig; 2nd Edition, 52-55) however, given that the**
**pig amnion does not form via cavitation as in humans but via folding, it is perhaps**
**unsurprising that amnion cells can be detected slightly earlier transcriptionally. Our**
**annotations of surface ectoderm vs amnion are also consistent with recent**
**descriptions of surface ectoderm and "late" primate amnion profiles (See**
**Rostovskaya et al., 2022; Cell Stem Cell, 29, 5, 744-759).**

*How can hypoblast be distinguished from the early definitive endoderm? From Extended*
*Figure 1 it appears that also hypoblast is expressing FOXA2, although at a lower level. It is*
*stated that FOXA2+/TBXT- cells emerge at E11.5, but in figure 5B they are present already*
*at E10.5. Also, in the in silico representation and quantification of the IF, there does not*
*appear to be any FOXA2+ cells displayed, please explain this difference.*

**Response: We apologize for the confusion about the FOXA2 shown in the E10.5. This**
**embryo was curved and the z plane shown in the original figure (blue line in the**
**picture below) included some hypoblast cells. We have captured a new z plane that**
**cuts through epiblast and emerging mesoderm, avoiding the underlying hypoblast**

(yellow line). We find no FOXA2 staining in epiblast cells consistent with the in-silico
data.

*Reviewer #3 (Remarks to the Author):*

*This is an important manuscript, with two parts that are well blend together. First there is a*
*careful comparative analysis of gene expression during gastrulation in four major*
*mammalian species. This will prove an extremely valuable resource. Then, as a derivative of*
*this, there is a study on the origin of the endoderm that provides incontrovertible evidence*
*that, in pig embryos, there is no common progenitor for endoderm and mesoderm and that*
*cells fated for this germ layers arise independently from the epiblast. It has an additional*
*point highlighting the origin of the node. There have been reports of a similar situation in*
*mouse embryos but, as pointed out by the authors, because of the topology and*
*developmental timing of gastrulation in the pig embryo, this study manages to provide very*
*sound evidence for what, in places has been debated in the mouse. After this manuscript,*
*this important feature of gastrulation can be considered a feature of mammalian*
*development. The statement is supported by detailed immunofluorescence analysis in*
*gastrulating pig embryos and also by functional experiments in pig and human embryonic*
*stem cells which, in addition, provide information about the signals mediating the fate choice.*

*The manuscript also makes a good decision of distinguishing between ‘primary’ and*
*‘secondary’ gastrulation which helps clarify the fate assignments discussed.*

*The manuscript will make an important contribution to our understanding of early stages of*
*mammalian development however, in places it is not clear and it would be helpful if the*
*authors could address some issues. In particular, it is important that they do not extrapolate*
*function from single cell analysis without empirical evidence.*

*There is one section I have problems with, namely the one called “organizer-like signalling*
*patterns of porcine cell types’. The heading is correct but then there are several issues*
*concerning references that should be corrected and also the authors should refrain from*
*making functional statements when what they are dealing with is single cell RNA seq data.*

*First, ref 45 might not be the best one for the point that in mouse it has been suggested that*
*the role of the ‘classical organizer’ is distributed between the node, the PS and the AVE (not*
*the hypoblast). This is the work of P. Tam and a direct reference might be more appropriate:*
*PMID 11566865*

**Response: We thank the reviewer for bringing this issue to our attention. We**
**apologise for this oversight and have added the primary references in lines 415-416.**

Also, in terms of references, neither 47 nor 48 refer to the phenotype of WNT3 mutants in
human; the reference is PMID: 14872406. Also, it is not clear to me that this phenotype is
similar to that of mouse Wnt3a mutants as these have a shortened trunk which is not the
case in the WNT3 mutants in human.

**Response: We apologize for the oversight; the references were misplaced during the**
**editing of the reference list (lines 419-423). We have also removed the mention of the**
**Wnt3a mouse phenotype.**

*In the discussion there is a statement: “our findings extend the understanding of the*
*emergence of the node in mammals, which contrary to chick models⁴⁵ and earlier DE*
*progenitors, we found no evidence for a role in A-P patterning”. It is not at all clear to me*
*how they can infer function in AP axis formation from the distribution of receptors in a UMAP.*
*The authors should be careful with these extrapolations. They should correct this.*

**Response: We agree that we cannot directly infer functionality from our data, thus we**
**have reworded this section of the discussion to reflect that transcriptional signatures**
**are inconsistent with a role in A-P patterning.**

*The expression of WNT3A in ED clusters is intriguing and important. Can they see the same*
*thing in the human data? is this the anterior? In the mouse it is expressed during primary*
*gastrulation in the anterior primitive streak and, later, in the Caudal Lateral Epiblast. It would*
*be helpful if they could clarify this.*

**Response: While this is indeed an intriguing idea, WNT3A expression is largely**
**absent from the human data (Tyser et al., 2021) so we cannot comment on whether**
**this is or is not the case.**

*When discussing the origin of the node cells, they make the statement (line 300 onwards)*
*“Given the APS origin of the node, and that all the cells in the APS/node trajectory do not*
*express markers of classical EMT (Fig. 4d) this also suggests that the FTS+ cells are not*
*mesendoderm”. This is intriguing and it appears as if these cells are closely associated with*
*the endoderm that challenges the general belief that they are ‘mesodermal’. Could the*
*authors comment on this? There are reports that the node has endodermal cells, but it*
*appears from this data that it is in the same transcriptional trajectory as the endoderm.*

**Response: We agree that this is an intriguing finding, transcriptionally the**
**Node/Notochord appears to share transcriptional features with endoderm such as the**
**expression of FOXA2, persistent expression of epithelial markers such as EPCAM**
**OCLN, KRT8 and KRT18. We have added to the latest manuscript version (Fig. 5d)**
**epiblast, DE and mesoderm module scoring of epiblast, DE, node and nascent**
**mesoderm clusters. We find that while node cells exhibited a significantly lower**
**endodermal score compared to DE itself it was higher than both epiblast and nascent**
**mesodermal clusters. By contrast, the mesodermal score differences between DE and**
**node clusters were not significantly different, suggesting the node is more**
**transcriptionally similar to DE than mesoderm (See lines 293-310). However, we also**
**note the reduced expression of the epithelial markers CLDN4 and FN1 in node**
**compared to DE. Similarly, we observe in addition to TBXT and CDX1 expression,**
**increased expression of the mesenchymal marker ZEB2 which is also expressed in**
**nascent mesoderm (Fig. 4d). Given these differences from endoderm and we see**
**FOXA2+TBXT+ (node-like) cells spatially separated from FOXA2+TBXT- cells, as well**
**as a small number that appear to be ingressing adjacent to nascent mesodermal cells**
**(Fig 5d), this may suggest that node cells internalise via a discrete mechanism before**

**forming the notochordal process. We have also added a discussion of the origin of**
**the node in lines 538-550.**

*The experiments on the origin of the endoderm with pig and human embryonic stem cells*
*are convincing but it would be helpful if the authors could also use Wnt instead of Chiron as*
*there are reports that the response of the cells is different for both agonists of Wnt signalling.*

**Response: We performed an experiment using WNT3A in addition to ChiR, as**
**suggested. We found that WNT-only stimulation in human ESC induced robust FOXA2**
**and limited SOX17 after 48 hrs, however in pig EpiSC WNT alone induced modest**
**FOXA2 and SOX17, as with ChiR. Combination of WNT + ActA (100 ng/ml) resulted in**
**robust FOXA2 and SOX17 activation in human and pig. These results suggests that in**
**humans WNT3A induces a higher proportion of definitive endoderm (as defined by the**
**amount of FOXA2/SOX17 expression) compared to ChiR, which promotes mesoderm**
**(TBXT) and endoderm (Fig. 4). In the pig, WNT combined with ActA has a stronger**
**more efficiently induces DE differentiation compared to ChiR (1uM). We added this**
**data to the manuscript (lines 472-482).**

*It would also be helpful if they could discuss how their observations relate to the data on the*
*emergence of endoderm in hESCs and the reported role that Nodal and Wnt play on it and*
*which differs from what their results propose. They mention this in the text but never address*
*it*

**Response: We have added references to previous studies comparing the effects of**
**WNT3A and CHIR (Teo et al 2014, Massey et al., 2019). Despite differences in**
**experimental set-up, we found that WNT3A is more efficient than CHIR inducing DE**
**differentiation from human and pig stem cells, albeit with different sensitivity between**
**species (lines 472-482).**

*Finally, an important event in 'secondary' gastrulation is the appearance of the Caudal*
*Lateral Epiblast. This is a population characterized by the appearance of the node (as they*
*point out) and also of a population of cells coexpressing TBXT, TBX6 and SOX2. It would be*
*helpful if the authors could pinpoint this in their data and highlight this moment in the*
*comparative map.*

**Response: We agree with the reviewer that while this population is interesting and**
**would serve as a useful landmark for secondary gastrulation. We have referred to**
**these node cells in line 228. With regards to the TBXT/TBX6/SOX2 population, classic**
**marker of NMPs, were not annotated as the version of the pig genome used as a**
**reference does not contain the SOX2 gene, preventing us of making a conclusive**
**claim. Reference-query mapping of mouse annotations to our pig single-cell data (Fig.**
**2a) suggests that these cells likely exist within our dataset however, without the SOX2**
**gene defining this population we decided not to refer to these cells in our study.**

REVIEWER COMMENTS

Reviewer #1 (Remarks to the Author):

I experienced some confusion while reviewing the rebuttal due to the absence of updated Extended Data figures, which hindered my ability to assess the responses to concerns 4, 7, and others. While many of my major and minor concerns have been addressed satisfactorily (3, 5, 8, 9, 11 and 1, 2, 3, 5, 6, 7, 8, 11 respectively), there remain issues that I believe need further clarification or resolution.

Major concern 1: My initial concern pertained to the potential existence of pig-specific cell populations that could render previous label transfers inaccurate. The introduction of a "no match" category in the label transferring process is reassuring. However, I recommend that the authors provide a more detailed description of these "no match" cell types in pigs and elucidate why these cells do not have counterparts in mice. An analogous explanation for the absence of certain mouse/monkey cell types in pigs would also be beneficial.

Major concern 2: Contrary to the authors' assertion of the impracticality of aligning different species due to their varied gastrulation stages, I believe there exist methodologies capable of facilitating such comparisons. The authors could examine the expression dynamics of marker genes like *FOXI*, *FOXA*, and *HAND* across mouse, monkey, and pig lineages. Utilizing the dynamic time warping algorithm might offer a novel approach to aligning these developmental processes.

Major concern 4: The rebuttal's Figure 1 demonstrates that different normalization methods yield divergent outcomes. This observation casts doubt on the reliability of the drawn conclusions. A more robust defense of the chosen method or consideration of alternative approaches is warranted.

Major concern 6: The absence of intermediate cell states might be due to a lack of detailed, second-round cell type identification. I recommend focusing on border cells between APS and DE, manually separating them, and analyzing their EMT gene expressions. Additionally, Figure 4d should include a column indicating time information in the heatmap.

Major concern 7: While I understand the rationale behind using different classification standards in various analyses, I strongly suggest the authors manually refine these standards to ensure consistency across the study.

Major concern 10: Figure 5a's omission of pseudotime or actual time data detracts from its clarity. The significance of several branching points in the early caudal epiblast warrants further investigation. Furthermore, discrepancies between the cell types depicted in Figure 5a and their corresponding label annotations, such as cluster 7, need resolution.

Minor concern 4: Given the identified inaccuracies in the human/mouse cell type definitions, a revised cell map for these species would be highly beneficial.

Minor concern 9: The authors should specify whether the endoderm populations were defined through a single comprehensive analysis or multiple independent ones. If the latter, I recommend manual adjustment of the standards to avoid inconsistencies.

Minor concern 10, The link between the scarcity of node cells identified before E12.5 and the node/notochord's primary involvement in secondary gastrulation is understandable. However, the premise that certain genes (NODAL, CER1, LEFTY2, DKK1) are exclusively produced by DE and not node-fated APS cells does not necessarily support this conclusion. This statement requires revision for accuracy.

Reviewer #2 (Remarks to the Author):

Authors have satisfactorily addressed all my comments.

Reviewer #3 (Remarks to the Author):

The authors have dealt satisfactorily with the issues I have raised, as well as with those of the other reviewers.

**Response to reviewer comments**

We thank the reviewers for their time in reviewing the revised version of our paper. Below (in
**bold**) we respond to the comments made by R1.

**Reviewer #1 (Remarks to the Author):**

*I experienced some confusion while reviewing the rebuttal due to the absence of updated*
*Extended Data figures, which hindered my ability to assess the responses to concerns 4, 7,*
*and others. While many of my major and minor concerns have been addressed satisfactorily*
*(3, 5, 8, 9, 11 and 1, 2, 3, 5, 6, 7, 8, 11 respectively), there remain issues that I believe need*
*further clarification or resolution.*

**We apologise that these were not given initially due to an error with the upload,**
**however the Extended data figures were provided following the online submission. We**
**believe these were shared with all the reviewers.**

*Major concern 1: My initial concern pertained to the potential existence of pig-specific cell*
*populations that could render previous label transfers inaccurate. The introduction of a "no*
*match" category in the label transferring process is reassuring.*

*However, I recommend that the authors provide a more detailed description of these "no*
*match" cell types in pigs and elucidate why these cells do not have counterparts in mice. An*
*analogous explanation for the absence of certain mouse/monkey cell types in pigs would also*
*be beneficial.*

**Response: As we indicated previously, the “No match” cells are not cells that exist in pig**
**but have no equivalent cell type mice, but rather they do not match an annotated cell**
**type in mice. These cells were initially called “NA” by the authors of Pijuan-Sala et al.,**
**and the reviewer objected to this label. These NA/No match cells appear to exist in all**
**three species; however, they have no definitive identity in that there are no known**
**marker genes that distinguish them as a particular cell type. These cells appear to have**
**been removed from the UMAPs by the authors of Pijuan-Sala et al. so they do not have**
**UMAP coordinates and are not visible in any of their figures, they can be seen as a**
**heterogeneous grouping of different cell types in Fig 2e near the allantois cluster.**
**Regarding their identity, we cannot be sure that they are not some sort of artefact**
**however in the case of our own study, and the studies conducted by Pijuan-Sala et al**
**and Zhai et al these cells were not identified as doublets, apoptotic cells or low-quality**
**cells; so while they cannot be excluded from the dataset, they do not appear to have a**
**distinct identity. Regarding the existence of pig-specific cell types, at the macro-level we**
**find that we have matches for all major cell types in all three species. While we agree**
**with the reviewer that the possibility of pig-specific subtypes (for example Rauber’s**
**layer present before gastrulation) may be an intriguing possibility this would require**
**extensive investigation and is beyond the scope of our current study.**

*Major concern 2: Contrary to the authors' assertion of the impracticality of aligning different*
*species due to their varied gastrulation stages, I believe there exist methodologies capable of*
*facilitating such comparisons.*

**Response:** As we stated previously, the comparisons between pig, mouse and monkey
were done between datasets that cover the start of gastrulation before the primitive
streak is visible via brightfield microscopy through to at least the 10-somite stage. These
datasets are as well-matched as possible. In response to the reviewer's concern we have
provided evidence of similar methodologies performed in several high-profile papers
that compare time course datasets, rather than pairwise comparisons between stage-
matched embryos (See Tyser et al., 2021, Nature, 600, 285-289; Zhai et al., 2023;
Nature, 612, 732-738; Ton et al., 2023, Nature Cell Biology, 1-12 as mentioned
previously, see also Mayshar et al., 2023, Cell, 186, 2610-2627) further many of these
studies explicitly show discrepancies in cell differentiation dynamics and/or composition
between similarly staged embryos of different species (for example the presence of
advanced blood cell types in human mid-gastrula embryo). Given that the reviewer has
not provided any specific references or examples of such methodologies that would
enable better matching, we are unable to take any action with this request.

*The authors could examine the expression dynamics of marker genes like FOXI, FOXA, and*
*HAND across mouse, monkey, and pig lineages. Utilizing the dynamic time warping*
*algorithm might offer a novel approach to aligning these developmental processes.*

**Response:** we feel that the addition of DTW analysis is not needed given our success in
integrating the datasets across time, and the other time specific breakdowns of the
integrated datasets presented within the manuscript.

*Major concern 4: The rebuttal's Figure 1 demonstrates that different normalization methods*
*yield divergent outcomes. This observation casts doubt on the reliability of the drawn*
*conclusions. A more robust defense of the chosen method or consideration of alternative*
*approaches is warranted.*

**Response:** We provided a detailed explanation in our previous rebuttal, including a
specific example (Fig. 1), demonstrating that the reviewer's suggested method of scaling
would result in inconsistencies between the raw count data, normalised data and scaled
data. In contrast, our method, which is outlined by the developers of Seurat, results in
no such inconsistencies. Indeed, numerous papers have utilised the same methodology
(The paper "Comprehensive integration of single-cell data", Stuart et al., 2019, Cell,
177, 1888-1902, which details the methodology for integrating datasets in Seurat V3 has
been cited by over 10,000 papers). We have also explained that differential gene
expression is performed on the normalised data, so using the reviewer's suggested
method would result in visualisations conflicting with the differential expression
calculations. Given the large number of papers utilising our chosen methodology and
the fact that the reviewer offers no justification or examples of papers using their
suggested methodology, we believe we have offered a satisfactory explanation justifying
the methods applied to our datasets.

*Major concern 6: The absence of intermediate cell states might be due to a lack of detailed,*
*second-round cell type identification. I recommend focusing on border cells between APS and*
*DE, manually separating them, and analyzing their EMT gene expressions.*

**Response:** We thank the reviewer for their suggestion. We do agree with the reviewer
that the dynamics of EMT-related genes do offer some value. So, in line with the
reviewer's suggestion, we have plotted endoderm and mesoderm fated cells along with

the earliest cells from DE and nascent mesoderm clusters ordered by their pseudo time
values in Supplementary Data Figs 3-6 allowing the expression of “border cells” to be
viewed. While interesting, we find that this does not alter our previous conclusions and
given the extensive space taken up by these figures we have kept these in the
Supplementary Data rather than in the main figures. Our analysis demonstrates there
is a distinct transition between epiblast, APS and DE states and indeed we see
intermediate cells between these states, however, we do not find any evidence of DE
progenitors undergoing “classical EMT”. In lines 239 to 260 we describe that the DE
progenitors maintain many aspects of their epithelial identity for instance expression of
EPCAM, CLDN6, KRT8 and KRT18 however we have also noted the expression of
several genes associated with EMT such as CDH2 and FN1, the latter which is more
highly expressed in DE. We suggest that the mechanism by which APS-DE
intermediates ingress differs significantly from that of PS-nascent mesoderm
intermediates. We have also reworded lines 239-240, 244-247 and 286-291 to ensure
clarity of this result.

*Additionally, Figure 4d should include a column indicating time information in the heatmap.*

**Response:** We agree with the reviewer that this is useful to present in the figure,
therefore, we added stage information to the heatmap of Fig 4d.

*Major concern 7: While I understand the rationale behind using different classification*
*standards in various analyses, I strongly suggest the authors manually refine these standards*
*to ensure consistency across the study.*

**Response:** While we understand the reviewer’s position we respectfully disagree. We
feel that our chosen methodology is the most appropriate and logical way of displaying
this complex dataset. Given that we have only identified cell subtypes of specific lineages
(for example somitic mesoderm, endoderm, epiblast, primitive streak etc) this has not
been done for every lineage (for example blood, trophoblast or PGCs, as it is beyond the
scope of this study) it would be inappropriate to display that data as such. The format of
identifying “macro-level” cell types followed by investigations of subtypes will be a
familiar and clear format for most readers of single-cell resource papers like this one.
Indeed, several high-profile papers have presented data this way including the papers
mentioned above (Pijuan-Sala et al., 2019, Nature, 566, 490-495; Tyser et al., 2021,
Nature, 600, 285-289; Zhai et al., 2023; Nature, 612, 732-738; Ton et al., 2023, Nature
Cell Biology, 1-12; Mayshar et al., 2023, Cell, 186, 2610-2627). To aid clarity we have
ensured that all UMAPs have now been labelled with the headings: Clusters and
subclusters. Further, we have included in the available metadata both cell type and cell
subtype columns so that if readers prefer to utilise cluster or subcluster information it is
readily available.

*Major concern 10: Figure 5a's omission of pseudotime or actual time data detracts from its*
*clarity. The significance of several branching points in the early caudal epiblast warrants*
*further investigation. Furthermore, discrepancies between the cell types depicted in Figure*
*5a and their corresponding label annotations, such as cluster 7, need resolution.*

**Response:** In this case, we do not feel the inclusion of pseudo time would be of much
benefit, primarily because it would indicate what is already visible from the UMAP
embeddings (for example that the nascent mesoderm and endoderm are the farthest

away from the epiblast and as such will be given higher pseudo time values, while
notochord will be given lower pseudo time values). In this case, we believe pseudo time
would simply reflect the transcriptional similarity of cells rather than giving any
insights into the order of the cell types' emergence and does not warrant inclusion in the
main figures. In case other readers feel similarly to the reviewer however we have
included a UMAP coloured by Monocle3 pseudo time in Supplementary Data Fig 3a.
We agree with the reviewer that real-time developmental timing information is useful,
we have already shown the presence of these cell types over time in Fig 4c. In response
to the reviewer's request, we have also included a UMAP coloured by the real-time
staging information in Supplementary Data Fig 3b. Regarding the several branching
points in the early caudal epiblast, it is to be expected that not all of the cells will be
transitioning toward mesodermal or APS/DE fates. These trajectories may reflect cell
division or transition toward a late caudal epiblast/NMP fate. While we agree with the
reviewer that this dataset will be a useful resource for investigating numerous other
cellular fate decisions in pigs/mammals, our focus was on the factors impacting
mesodermal vs APS/Endoderm fate decisions and thus further explorations are beyond
the scope of our current study.

*Minor concern 4: Given the identified inaccuracies in the human/mouse cell type definitions,*
*a revised cell map for these species would be highly beneficial.*

**Response: We agree with the reviewer's comment that these would be useful inclusions.**
**We have a revised cell map for these species in Fig. 2e and we have added an additional**
**figure in Extended Data Fig.6c showing human cells with pig cell type annotations.**

*Minor concern 9: The authors should specify whether the endoderm populations were defined*
*through a single comprehensive analysis or multiple independent ones. If the latter, I*
*recommend manual adjustment of the standards to avoid inconsistencies.*

**Response: This analysis was performed in a similar manner to the analyses shown in**
**Fig.4, 5 and Extended Data Fig.7. In brief, endodermal clusters (both embryonic and**
**extra-embryonic clusters were extracted from the main dataset and sub-clustered using**
**a new clustering resolution of 0.5. We then used a list of literature-curated markers for**
**cell-subtype identification. Regarding the adjustment of clustering standards, please see**
**the above response on line 131 within this document.**

*Minor concern 10, The link between the scarcity of node cells identified before E12.5 and the*
*node/notochord's primary involvement in secondary gastrulation is understandable.*
*However, the premise that certain genes (NODAL, CER1, LEFTY2, DKK1) are exclusively*
*produced by DE and not node-fated APS cells does not necessarily support this conclusion.*
*This statement requires revision for accuracy.*

**Response:**
**We have rephrased this sentence as requested on lines 438 to 440.**

REVIEWERS' COMMENTS

Reviewer #1 (Remarks to the Author):

Although the authors have addressed most of my concerns, I remain concerned about the cross-species analysis section. The unusual arrangement of apparently different cellular identities between pig and mouse not only reflects the asynchrony of developmental processes, but may also be a conceptual confusion resulting from mislabeling in this or previous studies, and should therefore be carefully distinguished and discussed. At least, the authors should discuss this conceptual issue in the discussion section

**Response to reviewer comments**

We thank the reviewer for their comment on the revised version of our paper. Below
(in **bold**) we indicate how we address this point in the final version of the ms.

**Reviewer #1 (Remarks to the Author):**

*Although the authors have addressed most of my concerns, I remain concerned*
*about the cross-species analysis section. The unusual arrangement of apparently*
*different cellular identities between pig and mouse not only reflects the asynchrony*
*of developmental processes, but may also be a conceptual confusion resulting from*
*mislabeled in this or previous studies, and should therefore be carefully*
*distinguished and discussed. At least, the authors should discuss this conceptual*
*issue in the discussion section.*

As we indicated in our previous rebuttal when finalising our annotations we have
considered studies in mouse, monkey and human and generated Fig. 2e and suppl.
Fig 6c. We have a fully referenced list of all the markers used for cell annotations
and we used reference mapping to identify transcriptionally similar cells in the other
species prior to any comparisons. This means that only cells that have broad
transcriptional similarities were compared. This is the first study to compare a closely
aligned single-cell map of the entire gastrulation period across mammals. We believe
this will be a point of reference for future cross species investigations. To reflect the
reviewer concern, we have added an additional sentence in the first part of the
discussion to highlight the need to for caution when comparing species (See lines
522-527). We have also added a more detailed section discussing limitations of
cross-species comparisons (See lines 588-605).
